# Data encoding for healthcare data democratization and information leakage prevention

Anshul Thakur [1] ✉, Tingting Zhu [1,4], Vinayak Abrol[2,4], Jacob Armstrong[1], Yujiang Wang [1,3] ✉ & David A. Clifton[1,3]

The lack of data democratization and information leakage from trained models hinder the development and acceptance of robust deep learning-based healthcare solutions. This paper argues that irreversible data encoding can provide an effective solution to achieve data democratization without violating the privacy constraints imposed on healthcare data and clinical models. An ideal encoding framework transforms the data into a new space where it is imperceptible to a manual or computational inspection. However, encoded data should preserve the semantics of the original data such that deep learning models can be trained effectively. This paper hypothesizes the characteristics of the desired encoding framework and then exploits random projections and random quantum encoding to realize this framework for dense and longitudinal or time-series data. Experimental evaluation highlights that models trained on encoded time-series data effectively uphold the information bottleneck principle and hence, exhibit lesser information leakage from trained models.

In recent years, deep learning has demonstrated remarkable success in a wide variety of fields[1], and it is expected to have a significant impact on healthcare as well[2]. Many attempts have been made to achieve this breakthrough in healthcare informatics, which often deals with noisy, heterogeneous, and non-standardized electronic health records (EHRs)[3]. However, most clinical deep-learning tools are either not robust enough or have not been tested in real-world scenarios[4,5]. Deep learning solutions, approved by regulatory bodies, are less common in healthcare informatics, which shows that deep learning hasn't had the same level of success as in other fields such as speech and image processing[6]. Along with well-known explainability challenges in deep-learning models[7], the lack of data democratization[8] and latent information leakage (information leakage from trained models)[9,10] can also be regarded as a major hindrance in the development and acceptance of robust clinical deep-learning solutions.

In the current context, data democratization can be described as making digital healthcare data available to a wider cohort of AI researchers. Achieving healthcare data democratization can result in global clinical models that are trained on data sampled from multiple geographical locations instead of being limited to a single site. These models are expected to be robust to population-specific distribution shifts and to exhibit better generalization. The wider access to healthcare data might also facilitate algorithmic contributions tailored for healthcare applications through a broader AI research base. However, healthcare data is sensitive and is rightly protected by data privacy laws making data democratization difficult[11,12].

On the other hand, latent information leakage is referred to as learning the non-targeted latent information about the underlying training population[10]. Higher modeling complexity of deep-learning models often facilitates the learning of this non-targeted information

[1]Department of Engineering Science, University of Oxford, OX3 7DQ Oxfordshire, UK. [2]Infosys Centre for AI, IIIT Delhi, Delhi, India. [3]Oxford Suzhou Centre for Advanced Research, Suzhou, China. [4]These authors contributed equally: Tingting Zhu, Vinayak Abrol. ✉e-mail: anshul.thakur@eng.ox.ac.uk; yujiang.wang@oscar.ox.ac.uk

that may act as an inductive bias to improve the predictive performance of models. However, the latent information can be sensitive or help in inferring the information such as age, sex, and chronic or acute medical conditions of the patients. The revelation of this sensitive patient information can be considered a privacy violation. Hence, data democratization and prevention of latent information leakage are two of the important factors required to develop better clinical deep-learning solutions that are secure and widely acceptable.

Data democratization can be equated with the irreversible de-identification of healthcare data so that no patient can be linked to an electronic health record (EHR). A truly de-identified dataset cannot be considered sensitive or private, so sharing it publicly would not result in a violation of any data privacy laws[13]. However, researchers have not developed a truly irreversible de-identification mechanism, and there is always a risk of re-identification[11,13,14]. It is a common practice to anonymize healthcare data, but the resulting data might not always be considered to be completely de-identified. In general, the notion of anonymity or de-identification is closely related to the amount of computational effort and time required to re-identify a patient from the data. An EHR can be considered non-anonymous (even after the anonymization process) if the efforts to re-identify the patient are considered reasonable. The reasonable efforts are subjective and should often change with advancements in technology[11]. As a result, simple data anonymization is not enough to achieve true de-identification and data democratization. Hence, there is a requirement for information-processing mechanisms that could mask private information while retaining the data semantics to enable data sharing or democratization.

Aside from data democratization, trained clinical deep-learning models also raise privacy concerns. These models have been shown to learn biomarkers of diabetic retinopathy, anemia, and chronic kidney disease from fundus images[15]. Apart from that, deep-learning models can also predict gender, sex, ethnicity, and smoking status from a fundus image[16]. Hence, it is quite possible that a model trained for predicting diabetic retinopathy from fundus images can learn a feature representation that may reveal non-targeted patient characteristics and sensitive information regarding the ailment of a patient suffering from chronic kidney disease and anemia. In the same way, a model trained for mortality prediction based on the first 48 hours of hospitalization in the intensive care unit (ICU) can provide information on the patient's acute as well as chronic conditions that may or may not be related to the current ICU stay or mortality prediction (see Results). The extensive feature extraction in deep-learning models results in better performance for the targeted task and the discovery of new non-targeted or passive digital biomarkers for various diseases, thereby improving healthcare provision. This disclosure of non-targeted information, however, violates the privacy of the patients and poses an ethical dilemma.

Deep learning models can be seen as a combination of feature extraction layers mapping an input example to a compressed, semantic representation or embedding and the last classification layer mapping the embedding to the model output or predictions (Fig. 1d). According to the information bottleneck (IB) principle, an ideal model should minimize mutual information between input and embedding while maximizing it between embedding and the model output[17,18]. In other words, the embedding extracted by the model should only contain task-specific information and must strip spurious or non-task-related information that might be present in the input. To avoid latent information leakage, clinical deep-learning models should be designed or trained to follow the IB principle and must only extract the relevant information from the input patient data.

This paper argues that encoding healthcare data can simultaneously achieve data democratization and prevent latent information leakage. To accomplish this, we envision an encoding framework that transforms pre-processed and anonymized longitudinal health records or multivariate time-series data into a new space. This envisioned encoding framework is characterized by one-way data transformations, imperceptibility of the encoded data, and preservation of semantic properties in the encoded data. A one-way transformation denotes the computational impracticality of recovering the original data from its encoded version. The imperceptibility of the encoded data refers to the inability to infer any information about the original data just by performing a simple manual or computational analysis of its encoded version. Feature scaling or normalization, for example, cannot be considered a viable method of encoding information. Finally, semantic preservation refers to the requirement that encoded data must preserve the semantic characteristics of the original data to an extent so that deep-learning models can be trained effectively over the encoded

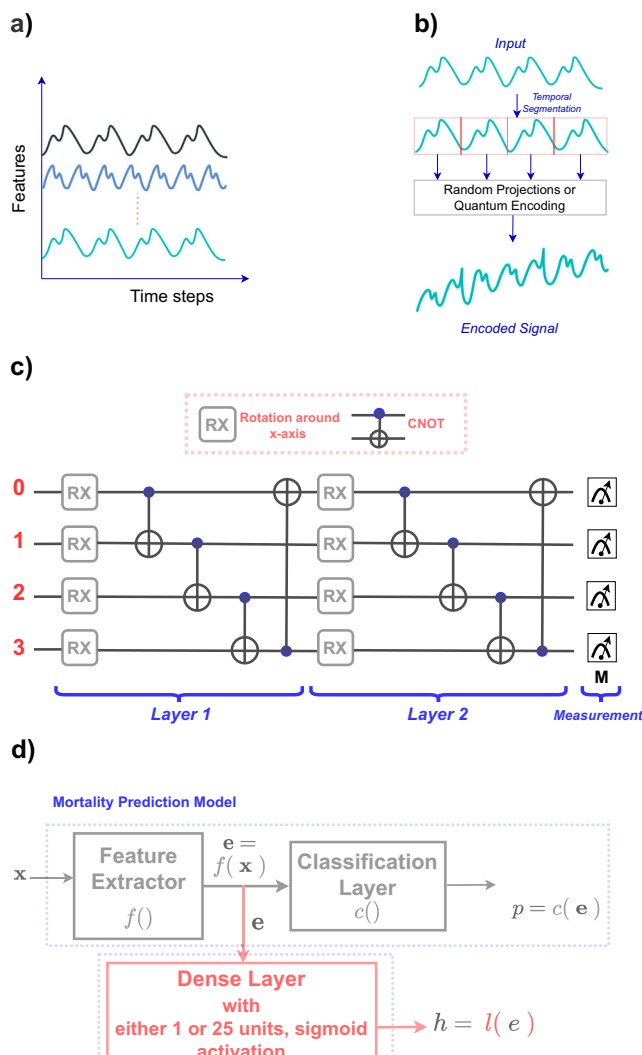

**Fig. 1 | A schematic illustration depicting the proposed encoding framework and its various components. a** Conceptual rendition of a multivariate time-series as a collection of multiple 1-d signals. **b** Illustration of the process of encoding one of the 1-d signals within a time-series using the proposed encoding framework. **c** Illustration of a quantum circuit that is composed of four wires, unitary rotation gates, and controlled-NOT (CNOT) gates. **d** Illustration of the setup used for evaluating the latent information leakage from the trained mortality prediction models. Penultimate layer embedding from the trained mortality prediction models is given as input to a linear or dense layer dealing with either gender or patient disorders predictions.

**Table 1 | Characteristics of MIMIC-III, PhysioNet, and eICU datasets**

| Dataset | # ICU stays | # Positive cases | Feature dimensions | Time-steps | # Train, validation, and held-out examples |
|---|---|---|---|---|---|
| MIMIC-III | 21,156 | 2799 | 60 | 48 | 14,698, 3222, 3236 |
| PhysioNet | 8000 | 1122 | 44 | 48 | 5120, 1280, 1600 |
| eICU | 122,588 | 57,434 | 284 | 12 | 68,648, 17,163, 36,777 |

data. In theory, the performance of models based on original data and encoded data should be the same.

The realization of this envisioned framework will enable the sharing of encoded healthcare data without violating privacy constraints. Ideally, encoded data is imperceptible, and the encoding process is practically irreversible. Therefore, it is very unlikely that any sensitive patient information can be derived from encoded data by either a manual or computational inspection. Nevertheless, there is an obvious trade-off between the imperceptibility and semantic preservation requirements of the envisioned encoding framework. A better semantic preservation results in lesser imperceptibility and vice versa. As a result, the encoded data can be seen as a deformed version of the original data, and much higher computational effort is required to extract its semantic characteristics. This nature of encoded data results in inherent regularization during model training and indirectly enforces the IB principle (see Results) to prevent latent information leakage.

This paper exploits random projections[19,20] and random quantum circuits[21,22] as information-processing tools to achieve the desired encoding framework for the multivariate time-series data. Both random quantum circuits and random projections can deform or project the data to a space where it becomes imperceptible. By exploiting random projections or random quantum circuits, the proposed encoding framework performs piece-wise or segment-wise temporal encoding of each feature or each 1-d signal of a multivariate time-series (Fig. 1b). Since there is no interference among features or signals of the original time-series, the resulting encoded time-series retains its semantic characteristics. However, random transformations deform each segment of a signal to make them incomprehensible. Due to the fact that the original data, encoding method, transformation matrix (used for random projections), and random quantum circuit will not be made public, it is extremely difficult to reverse the encoding process. Hence, data democratization can be achieved by sharing this encoded data among deep-learning researchers. Additionally, higher model complexity is required to extract the relevant semantic information from the deformed or encoded data resulting in regularization and thus enforcing the IB principle.

## Results
### Designed experiments for the performance evaluation
The proposed encoding framework is evaluated using publicly available datasets: 1) PhysioNet 2012 challenge[23], 2) MIMIC-III[24,25] and 3) eICU-CRD[26,27]. Both PhysioNet and MIMIC-III deal with in-hospital mortality prediction based on the first 48 hours of ICU stay. Similarly, eICU-CRD is used for the task of acute respiratory failure (ARF) prediction based on the first 12 hours of ICU stay. Each ICU stay is represented by a time-series with 48 and 12-time-steps (separated by 1 hour) for mortality and ARF prediction, respectively. Each step is represented by a 44, 60, and 284-dimensional feature vector in PhysioNet, MIMIC-III, and eICU datasets, respectively. Table 1 documents the total number of ICU stays or examples available in each dataset. In addition to the clinical features and task labels, meta-data about the patients corresponding to ICU stays are also available. This includes gender information in all datasets as well as chronic, acute, and mixed conditions afflicting patients in MIMIC-III and information about the ethnicity of the patients in eICU. More details about the clinical features representing time-series in all datasets can be found in Supplementary Notes 1, 2, and 3.

On the original as well as on the encoded data, we train 5 different neural networks on each dataset and compare their relative performances. These models include long short-term memory (LSTM)[28], temporal 1-D convolutions[29], multi-resolution temporal convolutions[30], transformer[31], and vision transformer[32]. More details can be found in the Section "Methods". To assess the latent information leakage from the trained models, a single dense layer mapping the penultimate layer embedding to the patient information is used. For the MIMIC-III dataset, gender and 25 latent or non-targeted patient disorders (acute, chronic, and mixed) are predicted from the penultimate layer embedding of the trained mortality prediction models. For PhysioNet, we only predict gender as the latent information. Similarly, we predict the gender and ethnicity of patients from the trained ARF prediction models. Since we are employing only a single linear layer to map embedding to either sex, ethnicity, or patient disorders (Fig. 1d), no further feature transformations are employed. The performance of this latent information prediction depends entirely on the nature of embedding. More details about this experimental setup can be found in the Section "Methods".

Apart from that, we also trained models on both original/raw and encoded datasets to predict the gender and ethnicity of the patients. The model architectures used for mortality and ARF prediction are also used for these prediction tasks.

### Performance on the encoded time-series data
The performance of various models on both the encoded and original datasets is illustrated in Fig. 2. Across all datasets, models trained and evaluated on the original data consistently outperform those dealing with the encoded time-series data. Specifically, concerning the MIMIC-III dataset, random quantum encoding, and random projection-based encoding resulted in an average relative performance drop of 3.52 (±1.25)% and 15.29 (±2.51)%, respectively. A similar trend was observed in the PhysioNet dataset, with an average relative performance drop of 5.13 (±1.94)% and 22.44 (±4.75)%. Likewise, the eICU dataset exhibited a drop of 2.13 (±1.59)% and 12.45 (±2.29)%. This decline is expected, considering that data encoding distorts the time-series to preserve patient information.

Despite the performance drop seen in models trained on the encoded data, particularly those using quantum encoding, these models appear effective in executing the target task. This suggests that the encoding framework, whether utilizing random projection or random quantum encoding, can maintain essential semantic characteristics in the deformed encoded data. Notably, random quantum encoding consistently outperforms random projections across all models and datasets, indicating that quantum encoding better preserves semantic characteristics while deforming the data through random quantum operations.

### Latent information leakage from the trained models
The performance for the task of predicting a patient's gender from the trained mortality and ARF prediction models is depicted in Fig. 3. According to the analysis of Fig. 3, we can effectively predict patients' gender from the trained models on original or non-encoded data. The behavior is common across all datasets and all models regardless of their modeling capacity. Similarly, the analysis of Fig. 4 illustrates that we can identify the patients' gender from the ARF models trained on the original time-series data. Although gender and ethnicity are not

sensitive information, these results highlight that trained models can indeed reveal the latent non-targeted patient characteristics.

Figure 5a illustrates the performance of predicting the patient disorders from the trained MIMIC-III models in a latent manner. The analysis of this figure highlights that all models trained on the original data generate representations or embedding that reveal information regarding the patients' disorders. Across all models trained on original data, a macro AUROC of approx. 0.7 is observed for the latent disorder prediction. It should be noted that the macro AUROC obtained by different models within this experiment is comparable to the performance achieved by the targeted patient phenotype prediction models (see Supplementary Fig. S1 of Supplementary Note 6). This shows that mortality prediction models are susceptible to leaking the patients' private medical information.

Figure 5b, c depict the performance of predicting the chronic and acute disorders (a subset of 25 disorders) from the trained LSTM mortality prediction models. Similar behavior is observed for all the other models considered in this study (see Supplementary Figs. S2 and S3 of Supplementary Note 7). The analysis of the figures shows that these models learn the characteristics that help infer or predict non-targeted patient disorders. We can predict both chronic and acute disorders that may or may not be correlated with the mortality prediction. According to the odds ratios[33] for these acute and chronic disorders (Supplementary Fig. S4 of Supplementary Note 8), most acute conditions exhibit a higher risk of mortality (odds ratio >> 1), while most chronic conditions are weakly associated with mortality (≈1). This shows that some conditions, such as shock and acute renal failure, are directly associated while others, such as chronic lipid

metabolism disorder and chronic renal disease, are not associated with mortality in the MIMIC-III patients corresponding to the ICU stays. Irrespective of odds ratios or the association between disorders and mortality, we can identify patients ailing from these ailments with an average AUROC of >0.7.

## Encoded data minimizes information leakage

The analysis of Figs. 3, 4, and 5 further highlights that the models trained on the encoded data exhibit lesser latent information leakage than the models trained on the original data. On average, MIMIC-III models trained on data encoded using quantum circuits and random projections (rather than original data) exhibited a relative drop of 20.11 (±2.45)% and 23.52 (±3.98)% in performance for the latent gender prediction task. The PhysioNet models also exhibited relative drops of 22.66 (±5.45)% and 28.21 (±8.98)% for the data encoded using the quantum circuit and the random projections, respectively. Similar behavior is observed for the eICU models where quantum encoding and random projection-based encoding resulted in a relative drop of 23.1 (±4.25)% and 31.11 (±7.6)% in the performance of the gender prediction task. The encoding data also resulted in a drop in the performance of the ethnicity prediction tasks. A similar trend is observed for the patient disorder prediction from MIMIC-III models. Quantum encoding and random projections resulted in a relative drop of 12.5 (±3.79)% and 18.75 (±5.45)% in the average macro AUROC score.

As discussed in Section "Introduction", models that follow the IB principle exhibit lesser information leakage. The drop in latent information leakage from models trained on the encoded data can be attributed to the lower mutual information (MI) between the model

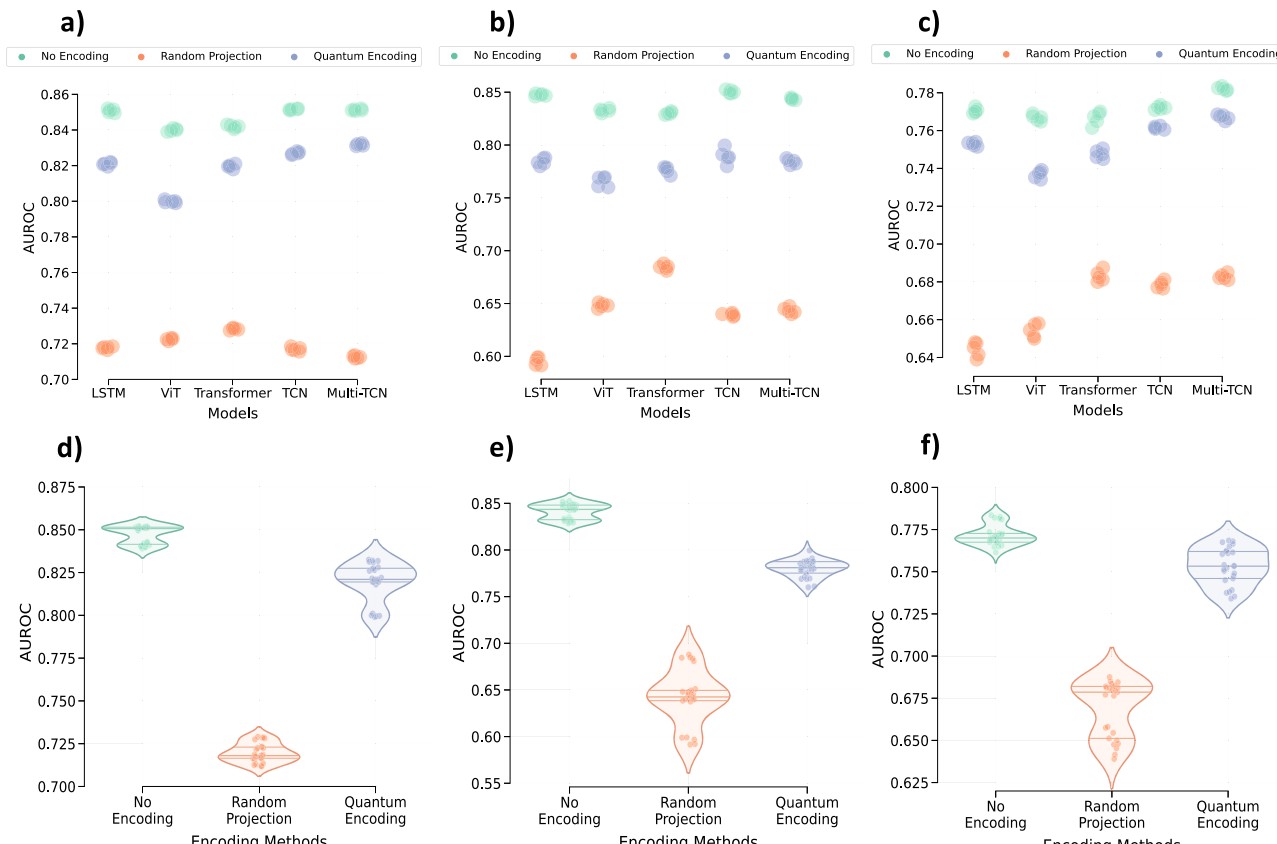

**Fig. 2 | Impact of the data encoding on the performance of different deep learning models.** Performance of LSTM, Vision Transformer (ViT), Transformer, Temporal Convolutional Network (TCN), and Multi-Branch Temporal Convolutional Network (Multi-TCN) on **a** MIMIC-III, **b** PhysioNet, and **c** eICU, respectively, obtained across five different runs. Violin plots illustrate the average performance of all models based on encoding methods for **d** MIMIC-III, **e** PhysioNet, and **f** eICU, respectively. The middle line within each violin plot represents the median, while the lines on either side represent the lower and upper quartiles. Source data are provided as a Source Data file.

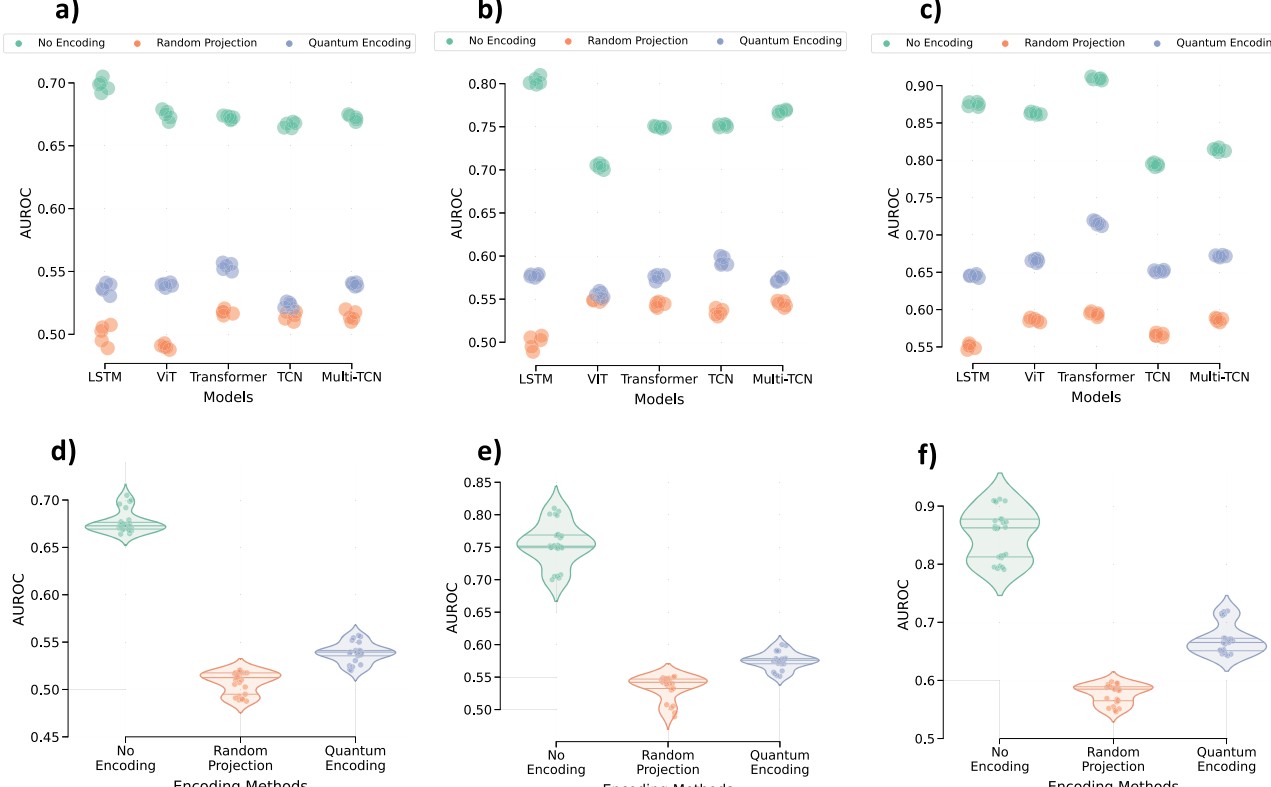

**Fig. 3 | The extent to which data encoding prevents the leakage of gender information from trained models.** Gender prediction from the latent embeddings obtained from different models trained on **a** MIMIC-III, **b** PhysioNet, and **c** eICU datasets, respectively. Violin plots illustrate the average performance of all models as a function of the encoding method on **d** MIMIC-III, **e** PhysioNet, and **f** eICU datasets, respectively. Every point on all plots represents the respective model performance obtained during one of the five runs. The middle line within each violin plot represents the median, while the lines on either side represent the lower and upper quartiles. Source data are provided as a Source Data file.

input (i.e., original or encoded) time-series and the penultimate layer embedding generated from the trained models. To uphold this claim, we estimated MI between penultimate embeddings obtained from the trained LSTMs and the input time-series examples. For the feasibility of MI estimation, we used the average and vectorized form of the input time-series to compute MI. Figure 6 illustrates the distribution of estimated MI between the input and the penultimate embeddings. It is clear from this figure that the utilization of encoded data minimizes the MI between the model input and the learned representation. As a result, it can be inferred that training models with the encoded data inherently enforce the IB principle in the training process. Hence, the learned embedding only retains the information required to predict mortality while stripping away the non-essential or non-targeted patient information.

The above analysis shows that random projections-based encoding provides maximum prevention against latent information leakage. However, if we analyze Fig. 3 along with Fig. 2, it is also evident that random projection-based encoding results in a larger drop in the performance of the targeted task. On the other hand, random quantum encoding provides more balance between the performance of the targeted task and the prevention of information leakage.

**Visual inspection of the encoded data**
The visual differences between the original and the encoded examples from the PhysioNet dataset are illustrated in Fig. S6 of the supplementary information document. The analysis of this figure makes it clear that both temporal trends and distribution of features in the original and the encoded time-series examples are noticeably different.

To further analyze the impact of the encoding process on the time-series data, 50 original and encoded examples from the positive (mortality) class of the PhysioNet dataset were randomly selected and averaged to obtain the original and encoded summary time-series. Figure 7 depicts the behavior of four randomly chosen features from these summarized time-series. Again, the distribution of magnitude as well as temporal trends of the encoded features is different from the original time-series features. By mere visual inspection, it is near impossible to perceive any information from the encoded data (both quantum encoding and random projections). Similar behavior is observed for the other features. Hence, the encoding process provides an additional layer of privacy over the de-identified data and might push the community a step closer to achieving data democratization.

**Predicting gender and ethnicity from original and encoded datasets**
Supplementary Figs. S7 and S8, documented in Supplementary Note 11, illustrate how different models perform when trained to predict gender and ethnicity from the raw and encoded time-series data directly. As with the latent gender and ethnicity prediction tasks, the time-series encoding also results in a significant drop in the performance of models trained on encoded time-series samples for predicting gender and ethnicity. Across all models, random projection results in a relative drop of 26.03%, 32.5%, and 33.33%, respectively on MIMIC-III, PhysioNet, and the eICU gender prediction tasks. Similarly, quantum encoding results in average relative drops of 13.7%, 24.1%, and 22.9%, respectively. Similar trends are observed for ethnicity prediction tasks. The analysis of these results provides strong evidence that time-series encoding makes it hard to infer sensitive characteristics that can readily be extracted from the raw time-series data. If we

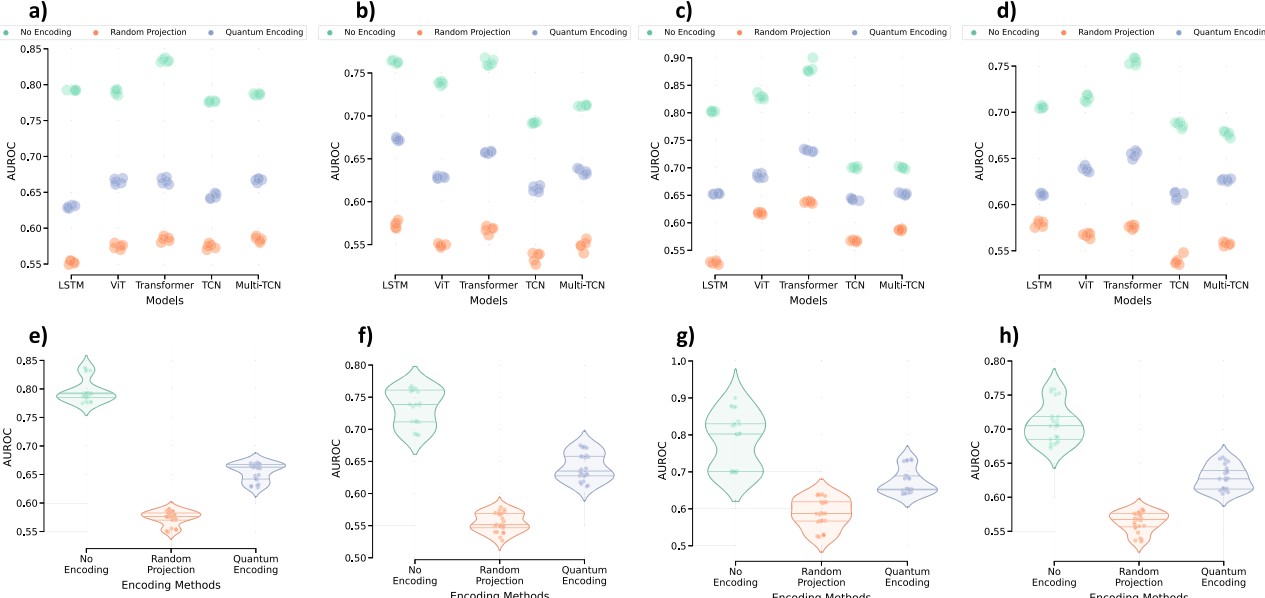

**Fig. 4 | The extent to which data encoding prevents the leakage of ethnicity information from the trained models.** Performance in the latent prediction of a patient's ethnicity as **a** Asian, **b** African-American, **c** Hispanic, or **d** Caucasian from various models trained on the eICU dataset, respectively. Similarly, violin plots illustrate the average performance across all models in predicting a patient's ethnicity as **e** Asian, **f** African-American, **g** Hispanic, or **h** Caucasian, respectively, based on encoding methods. Every point on all plots represents the respective model performance obtained during one of the five runs. The middle line within each violin plot represents the median, while the lines on either side represent the lower and upper quartiles. Source data are provided as a Source Data file.

analyze these results in association with mortality and ARF prediction tasks as well as latent prediction tasks, it is evident the proposed encoding framework achieves the desired characteristics of preserving semantics as well as masking sensitive information to a large extent.

### Data encoding and explainability

Encoded data is expected to retain semantic characteristics of the original data to a large extent such that models trained on original and encoded data exhibit similar behavior. Along with similar performance, the features relevant for predictions in models trained on both the original and the encoded data should largely be the same. While encoded data does retain semantic characteristics, there is a noticeable performance drop due to data encoding (Fig. 2). This shows that the behavior of models trained on encoded data could be different.

Shapely additive explanations (SHAP)[34] are employed on the LSTM models, trained using the original and encoded PhysioNet and MIMIC-III datasets, to study the impact of data encoding on the feature relevance. Figure 8 illustrates the top 10 relevant features identified by SHAP in each PhysioNet model. The analysis of this figure highlights that there is a huge overlap between the sets of relevant features identified for the original and the quantum-encoded models. Moreover, Glasgow comma score and blood urea nitrogen are regarded as the most relevant features in both models. Although there is some overlap between the relevant features of the original and the random projection-based encoded models, the overall behavior seems to be very different. Similar behavior is observed for the MIMIC-III models (see Supplementary Fig. S9 of Supplementary Note 12). Hence, it can be argued that random quantum encoding has been successful in retaining semantic characteristics such that the resultant models exhibit similar behavior to the original models up to an acceptable level.

### Discussion

This study proposes to encode the healthcare data to achieve data democratization and prevent information leakage. The irreversible and semantic preserving encoding framework outlined in this paper allows getting an imperceptible and deformed form of healthcare data

that can be shared among researchers without violating privacy constraints. Moreover, the inherent regularization imposed on neural network training due to the deformity of the training data is expected to induce the information bottleneck (IB) principle and potentially result in models that are less susceptible to latent information leakage (Fig. 6). The experimental results on three different time-series datasets and five different model architectures highlight that the proposed encoding framework achieves the desired behavior while outlining the potential of encoding frameworks for data democratization.

This paper explores random projections and random quantum operations to piece-wise encode the 1-d signals in a time-series as highlighted in Section "Methods" and Fig. 1. Compared to the original time-series signals, the resultant encoded signals exhibit different feature distributions and follow somewhat imperceptible trends (Fig. 7). Models trained on the encoded data perform well, highlighting that the semantics are effectively preserved (Fig. 2). Concomitantly, the information leakage from these models is significantly lesser than models trained on the original data (Figs. 3, 4, and 5). Thus, as desired, the proposed encoding framework results in encoded data that is visually imperceptible, effective for deep learning, and minimizes information leakage from the trained deep models.

Based on the performance comparison between models trained on data encoded using random projections and random quantum circuits (Figs. 2 and 3), it is evident that random quantum encoding balances the deformation of data and preservation of the semantic characteristics, which results in better models. Apart from the better performance of quantum encoding, retrieving the original data from its encoded version is theoretically harder as outputs of the quantum circuit or the state of qubits are observed by projecting them on a pre-defined basis state[35]. These measurements become the encoded signals, and estimating the qubit state from this measurement can be ambiguous as multiple qubit states could map to the same measurement. Even if the measurement weren't an issue, one would have to estimate the structure of quantum circuits (number of layers, number of gates, and nature of gates) as well as the parameters of rotation gates to reverse the encoding process possibly. In contrast, we only need to estimate the transformation matrix ($4 \times 4$) to reverse the

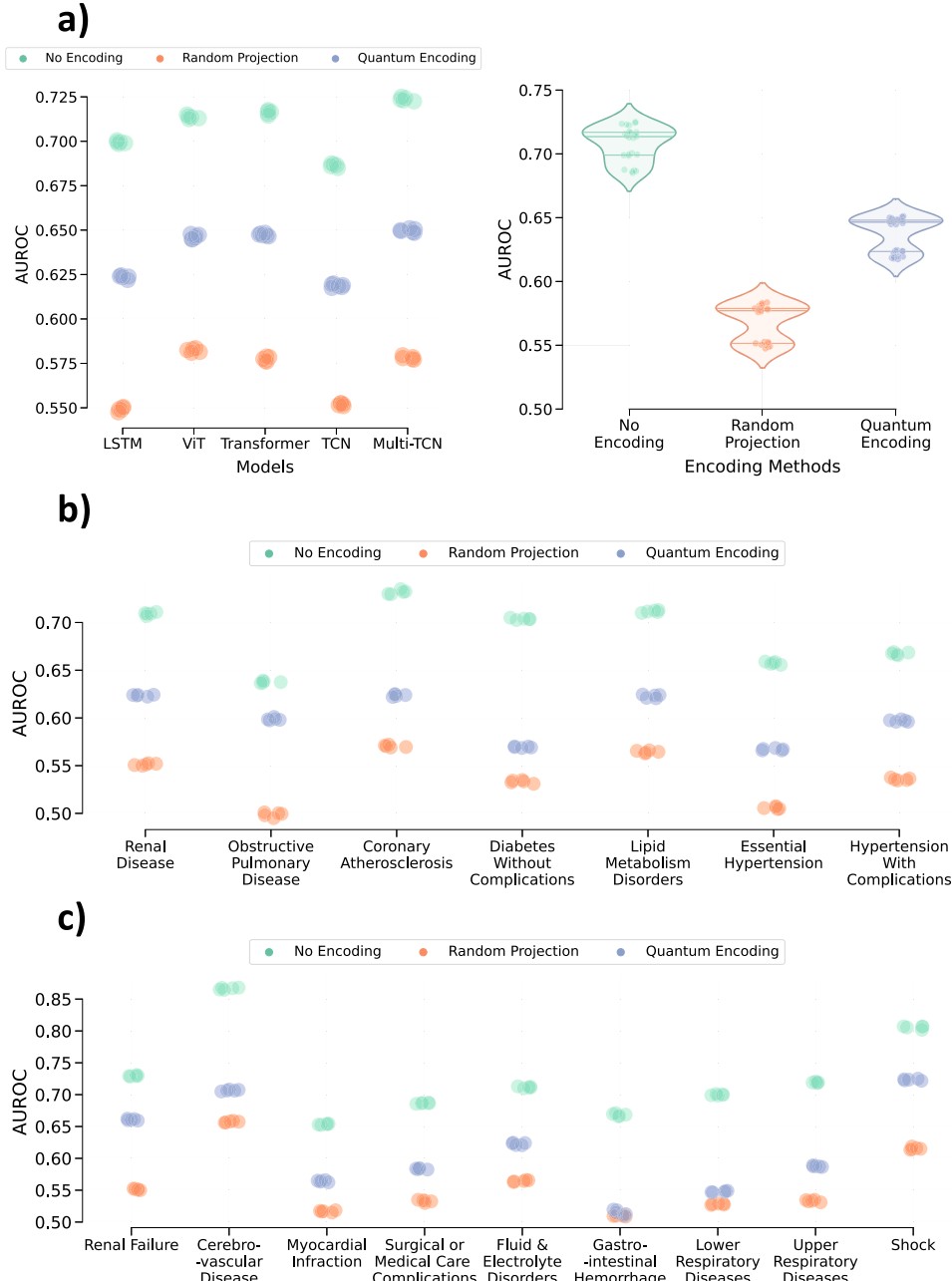

**Fig. 5 | The extent to which data encoding prevents the leakage of non-targeted patient conditions from trained patient-care models. a** Model-specific and average performance across all models for predicting 25 latent patient disorders using the penultimate embedding generated from models trained on the MIMIC-III dataset. The chronic and acute disorders shown in **b**, **c** are subsets of 25 different conditions considered in this work. A single model predicts the presence/absence of all 25 disorders. Every point on all plots represents the respective model performance obtained during one of the five runs. The middle line within each violin plot represents the median, while the lines on either side represent the lower and upper quartiles. Source data are provided as a Source Data file.

random projections or similar data transformations. It will be sufficient to have access to even one pair of original and encoded data to estimate this transformation matrix accurately. As a result, while random projection induces visible imperceptibility and preserves semantics to some extent, it cannot be considered an irreversible transform which is a major requirement of the proposed encoding framework. On the other hand, quantum encoding provides theoretical irreversibility while preserving semantics and inducing imperceptibility. Hence, it presents a better data transformation or encoding solution.

The encoding of data is also able to facilitate collaboration among multiple research entities without infringing upon the privacy of the patients. All data collection sites can potentially share their data among themselves so that every site can access the global data. As

discussed in Section "Introduction", the models trained on this global data are expected to be more generic and better at handling the population-specific distribution shifts. However, the random nature of encoding at each site will impede this cross-site collaboration. This problem can be solved by agreeing beforehand on the nature of data transformation, such as quantum circuit structure and rotation gate parameters. Thus, encoded data from each site will be in the same transformation space, allowing deep-learning models to be trained effectively. Similar to cross-site collaboration, federated learning also allows a central server to collaborate with multiple sites for training a global model without data sharing[36]. However, the structure of models is entirely decided by the server, and sites do not have any independence. Each site is expected to perform similar operations using its

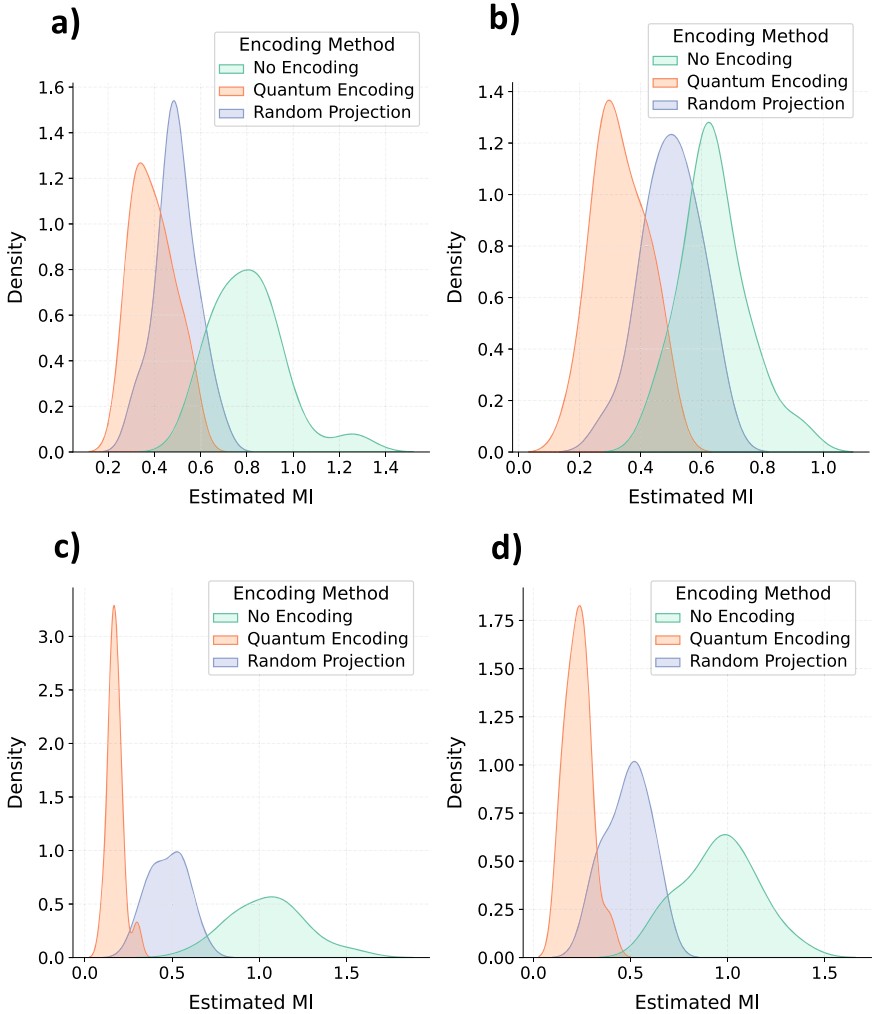

**Fig. 6 | Impact of data encoding on the information bottleneck.** Kernel density estimation plots depict the estimated mutual information (MI) between embeddings derived from trained LSTM models and the averaged input time-series in **a** MIMIC-III and **c** PhysioNet. Additionally, similar plots show the estimated MI between embeddings from the trained LSTM models and vectorized input time-series in **b** MIMIC-III and **d** PhysioNet. Source data are provided as a Source Data file.

local data. On the other hand, data encoding allows the researchers at each site to access the global data and work independently on any deep-learning algorithm.

As an alternative to data encoding, generative models such as generative adversarial networks have been used to generate data points that do not represent any real patients and theoretically can be shared publicly[37,38]. However, generative models capture the input distribution of the data points, and it is always possible to sample data points that are extremely similar to the input points or real patients. Similar to the subjectivity around the de-identification process (as discussed in Section "Introduction"), a sampled example that is similar to real patient data may or may not be considered a fabricated data point. Moreover, generative modeling requires extensive computational resources and a large amount of data to fabricate the data points effectively. On the other hand, the proposed encoding approach is an information-processing framework and does not require any training.

Upon reflection, this work reveals three shortcomings. Firstly, the proposed framework is designed to encode data for deep-learning models with advanced capabilities, hindering the utility of traditional machine-learning models with limited modeling complexities. Additionally, intentional disparities in summary statistics make statistical and epidemiological analyses unfeasible, limiting the utility of encoded to deep-learning applications. Secondly, both random projections and random quantum encoding lack a mechanism to control

deformation or balance imperceptibility and semantic information retention, leading to a performance drop in models trained on the encoded data. Finally, the proposed framework hasn't been evaluated on recent foundation models like TimeGPT-1[39]. These models are significantly larger, boasting extensive modeling capacities. Consequently, it is conceivable that these models may extract a wider range of non-targeted applications compared to the standard models assessed in this paper.

In the future, we will work towards inventing new non-linear or sub-linear data transformations that could either automatically balance the deformation and semantic retention trade-off or provide a hyper-parameter to control the degree of deformation in the encoded data, while being theoretically irreversible. Using such data transformations in the proposed encoding framework will improve the performance of the target tasks while enabling data democratization and preventing information leakage. Furthermore, future work will also deal with analyzing and evaluating foundation models on the encoded examples.

## Methods
### Proposed encoding framework
A uniformly sampled multivariate time-series is a collection of multiple 1-d signals representing features measured over time. Suppose $\mathbf{X} \in \mathbb{R}^{F \times T}$ is a time-series consisting of $F$ 1-d signals of length $T$, and $\mathbf{x} \in \mathbb{R}^T$

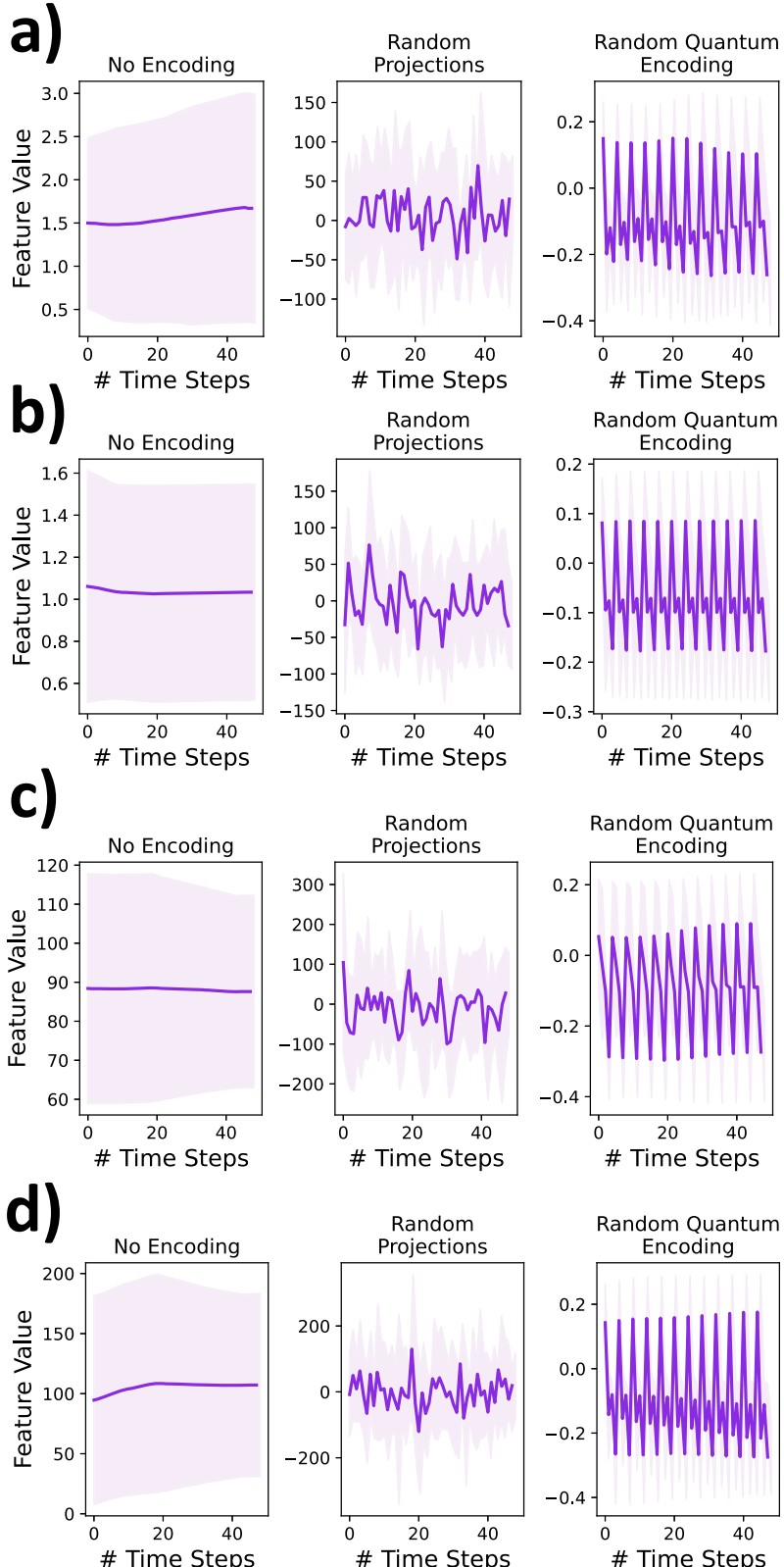

**Fig. 7 | Data encoding enhances imperceptibility.** The difference in average trends and the average magnitude of the original and encoded signals representing **a** cholesterol, **b** blood urea nitrogen, **c** alkaline phosphatase, and **d** alanine transaminase are examined. These signals are computed by averaging 50-time-series representing patients who eventually face mortality in the PhysioNet dataset. The shaded area surrounding the averaged signal represents the standard deviation. Source data are provided as a Source Data file.

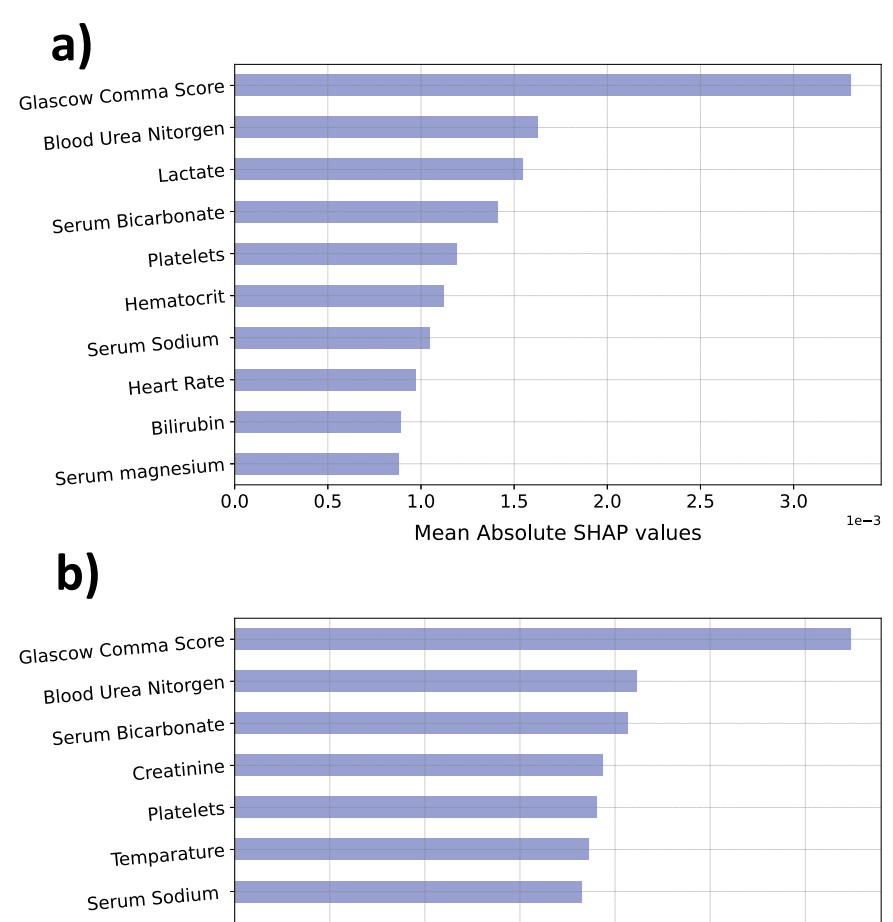

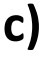

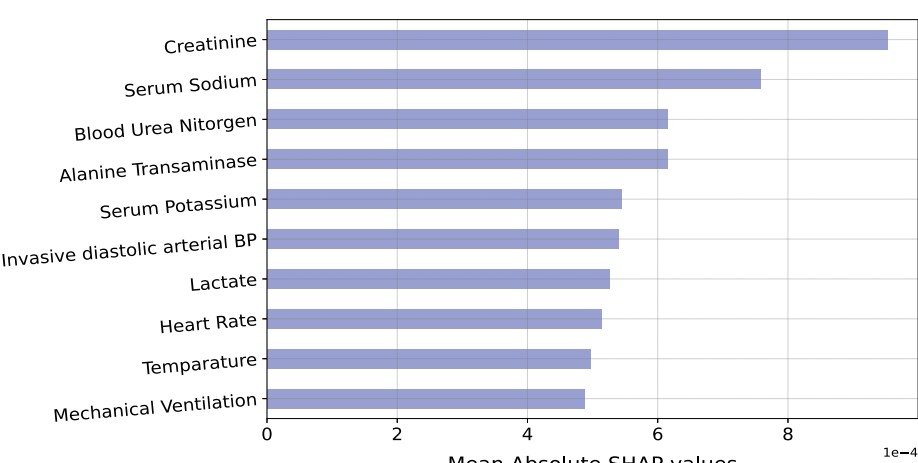

**Fig. 8 | Consistency in explainability of models trained on raw and the encoded data.** A comparison of SHAP-based feature importance in LSTM models trained on **a** original, **b** quantum encoded, and **c** randomly projected versions of the PhysioNet dataset. Source data are provided as a Source Data file.

or $\mathbf{x} = \left[ \mathbf{x}_1, \mathbf{x}_2, \mathbf{x}_3, \ldots \mathbf{x}_T \right]$ is one of the $F$ signals. The proposed framework transforms the time-series $\mathbf{X}$ by performing piece-wise encoding of every 1-d signal in $\mathbf{X}$. The framework divides the signal $\mathbf{x}$ into segments or chunks of length $n$ as $\hat{\mathbf{x}} = \left[ \mathbf{x}_{1:n}, \mathbf{x}_{n+1:2n}, \ldots \mathbf{x}_{(T-n+1):T} \right]$ and

applies transformation operation $f()$ on every segment:

$$\mathbf{e}_j = f(\hat{\mathbf{x}}_j) \ \forall \hat{\mathbf{x}}_j \in \hat{\mathbf{x}}, \tag{1}$$

where $\mathbf{e}_j \in \mathbb{R}^n$ is encoded version of $j$th segment of $\mathbf{x}$. Note that the dimensions of transformed/encoded and input segments are the same, and a segment length of $n = 4$ has been used across all experiments. Each encoded segment of length $n$ is temporally concatenated to obtain the encoded version, $\mathbf{e} \in \mathbb{R}^T$, of the signal $\mathbf{x}$ as: $\mathbf{e} = [\mathbf{e}_1, \mathbf{e}_2, \mathbf{e}_3 \ldots \mathbf{e}_{(T/n)}]$. Similarly, transformation or encoding operation is applied on all $F$ 1-d signals to transform $\mathbf{X}$ into the encoded time-series $\mathbf{E} \in \mathbb{R}^{F \times T}$. In this paper, we have used random projection and random quantum encoding as data transformation operation $f()$ in the proposed framework. Both these mechanisms are discussed below.

Random projection is a method of projecting the input data into a random subspace using a random projection matrix whose columns are of unit length[19,20]. It is mainly used for dimensionality reduction, and it approximately preserves the similarity among data points in the projected subspace as outlined by Johnson-Lindenstrauss lemma[40]. In this work, we are not interested in dimensionality reduction and are mainly concerned with projecting the input into a random subspace to make the data imperceptible. To attain this goal, we use a projection matrix $\mathbf{R} \in \mathbb{R}^{n \times n}$ whose entries are randomly sampled from Gaussian distribution $\mathcal{N}(0, 1/n)$. This projection matrix can be used to encode the $j$th segment $\hat{\mathbf{x}}_j \in \mathbb{R}^{n \times 1}$ of signal $\mathbf{x}$ as:

$$\mathbf{e}_j = \mathbf{R}\hat{\mathbf{x}}_j, \tag{2}$$

where $\mathbf{e}_j \in \mathbb{R}^{n \times 1}$ is the encoded version of the input segment. As discussed above, we have used a segment length of $n = 4$, so the projection matrix of $4 \times 4$ is used for data encoding.

Random quantum encoding refers to a process of data transformation through the use of a quantum circuit containing multiple gates with random parameters[21]. The quantum circuit used in this study is shown in Fig. 1c. This circuit is composed of the following components: qubits or wires, rotation gates, and controlled-not gates[41]. The circuit consists of four wires to represent four quantum bits or qubits. A qubit is a quantum system having a resting state $|0\rangle$ and an excited state $|1\rangle$. These states are mutually orthogonal and any qubit state $|\psi\rangle$ can be represented as a superposition of $|0\rangle$ and $|1\rangle$ as: $|\psi\rangle = a|0\rangle + b|1\rangle$, where $a$ and $b$ are complex numbers that must satisfy $|a|^2 + |b|^2 = 1$. $|a|^2$ and $|b|^2$ represent the probability of $|\psi\rangle$ being in $|0\rangle$ and $|1\rangle$, respectively. Initially, all four qubits are in a resting state. The number of wires or qubits is dictated by the length of the input segmented signal, i.e., $n = 4$. Secondly, rotation gates (RX) rotate the qubit around $x$-axis by $\phi_k$ (radians) on its Bloch sphere projection, where $k$ is the index of RX gate in the circuit. This rotation operator with $\phi_k$ randomly chosen parameters can be defined as:

$$RX(\phi_k) = \begin{bmatrix} \cos\frac{\phi_k}{2} & -\iota\sin\frac{\phi_k}{2} \\ -\iota\sin\frac{\phi_k}{2} & \cos\frac{\phi_k}{2} \end{bmatrix}. \tag{3}$$

The resultant qubit state $|\psi'\rangle$ after applying $k$th RX gate to qubit $|\psi\rangle$ is given as:

$$|\psi'\rangle = \begin{bmatrix} \cos\frac{\phi_k}{2} & -\iota\sin\frac{\phi_k}{2} \\ -\iota\sin\frac{\phi_k}{2} & \cos\frac{\phi_k}{2} \end{bmatrix} \begin{bmatrix} a \\ b \end{bmatrix}. \tag{4}$$

The final component, controlled-not (CNOT) gates, are used to entangle the two qubits and have no parameters. First qubit is considered as control and the second qubit is flipped if the control is $|1\rangle$. As we can see, CNOT deals with 2-qubit quantum system whose basis states are $\{|00\rangle, |01\rangle, |10\rangle, |11\rangle\}$. An input to CNOT gate is a linear superimposition of these basis states: $|\psi\rangle = a|00\rangle + b|01\rangle + c|10\rangle + d|11\rangle$, where $a, b, c$ and $d$ are the complex coefficients. Hence, CNOT operation can be defined as:

$$CNOT(|\psi\rangle) = a|00\rangle + b|01\rangle + d|10\rangle + c|11\rangle. \tag{5}$$

The whole quantum encoding process can be divided into three steps: (1) encoding input segment on wires, (2) processing qubits by quantum circuit, and (3) measuring the outputs. In the first step, the input segment $\hat{\mathbf{x}}_j$ is projected on wires of the circuit. Each element $(\hat{\mathbf{x}}_{j_n})$ of the input segment $\hat{\mathbf{x}}_j$ corresponds to $n$th wire or qubit. To encode the information from $\hat{\mathbf{x}}_{j_n}$ to $n$th qubit, we rotate this qubit by $\hat{\mathbf{x}}_{j_n}$ radians around the $y$-axis on its Bloch sphere projection. This rotation operator is described as:

$$RY(\phi_n) = \begin{bmatrix} \cos\frac{\phi_n}{2} & -\sin\frac{\phi_k}{2} \\ \sin\frac{\phi_n}{2} & \cos\frac{\phi_k}{2} \end{bmatrix}, \tag{6}$$

where $\phi_n$ is $\pi\hat{\mathbf{x}}_n^j$. The process of applying this operator is similar to RX gates (Equation 4). In the second step, after preparing the qubits as encoded versions of the input segment, these qubits are processed by the quantum circuit (Fig. 1c) described above. Finally, a measurement operation is performed to register the state of a qubit after applying all the quantum operations. In this work, we use the expectation of the Pauli-Z operator ($\mathbf{Z}$) to measure the output state of a qubit $|\psi\rangle$. We know that $\mathbf{Z}$ can be defined as[41]:

$$\mathbf{Z} = \begin{bmatrix} 1 & 0 \\ 0 & -1 \end{bmatrix}, \tag{7}$$

where $|0\rangle\langle 0| - |1\rangle\langle 1|$ is the spectral decomposition form of $\mathbf{Z}$. Then, the expected value of Pauli-Z operator for $|\psi\rangle$ can be determined as:

$$\langle\psi|\mathbf{Z}|\psi\rangle = \langle\psi|0\rangle\langle 0|\psi\rangle - \langle\psi|1\rangle\langle 1|\psi\rangle = |\langle 0|\psi\rangle|^2 - |\langle 1|\psi\rangle|^2. \tag{8}$$

Here $|\langle 0|\psi\rangle|^2$ and $|\langle 1|\psi\rangle|^2$ represents the probabilities of $|\psi\rangle$ being in states $|0\rangle$ and $|1\rangle$, respectively. Note that $\langle a|b\rangle$ represents the inner product between $|a\rangle$ and $|b\rangle$ in Hilbert space. For $n$th wire or qubit, the measured value ($e_{j_n}$) is regarded as the encoded version of the corresponding element $\hat{\mathbf{x}}_{j_n}$ of the input segment $\hat{\mathbf{x}}_j$. By considering all $n$ qubit measurements, we obtain an encoded version ($\mathbf{e}_j = [e_{j_1}, e_{j_2}, \ldots e_{j_n}]$) of the input segment $\hat{\mathbf{x}}_j$. The encoded signal $\mathbf{e}$ is obtained by temporally concatenating all the encoded segments $\mathbf{e}_j$.

## Models

This work has trained various neural network architectures for performing targeted and latent predictions. Firstly, the long short-term memory (LSTM) based model, previously used in mortality prediction[12], incorporates an LSTM with 256 recurrent units, followed by a linear layer with 1 node and sigmoid activation for binary prediction.

Temporal convolution neural networks, drawing inspiration from works such as refs. 29,30, leverage 1-dimensional convolution operations for time-series modeling. Our implementation features temporal convolutional networks with four temporal blocks followed by a linear layer with 1 node and sigmoid activation, mapping the 64-dimensional embedding to an output score. Each temporal block consists of two 1-dimensional convolution layers with 64 filters of size 9. Each convolution layer is followed by 1-dimensional batch normalization, parametric ReLU activation, and a dropout layer with a dropout probability of 0.75. Additionally, a multi-branch temporal convolutional network (Multi-TCN) is utilized, comprising two multi-branch temporal blocks followed by a linear layer with 1 node and sigmoid activation. Each multi-branch temporal block comprises three branches that process the input in parallel, with each branch featuring two 1-dimensional convolutional layers having 32 filters. The filters' kernel sizes in the branches are 5, 7, and 9, respectively. The last layer of the block is a 1-dimensional convolution layer with 96 filters of size 1, serving as an aggregator to select relevant features from all three branches.

Furthermore, transformer architectures, as introduced in[31], encompass an encoder and a decoder, each composed of multiple self-attention layers. A transformer encoder is utilized in this work, containing 1 attention layer with sixteen 256-dimensional heads, followed by two linear layers with 16 and F nodes. The output, shaped as T × F, where T is the time-steps and F is the feature dimensions, undergoes temporal pooling before being fed into a two-layered MLP classifier with 128 and 1 nodes for binary prediction. Additionally, the Vision Transformer (ViT), designed explicitly for images in[32], is employed for modeling time-series. The architecture mirrors that of the transformer, with the ViT featuring a learnable F-dimensional token appended to the input time-series. This token is then given as input to the MLP classifier instead of a temporally pooled representation, as is done in the classical transformer.

As previously mentioned, the models utilized are single-layer models, featuring either 1 node for latent binary prediction tasks or 25 nodes followed by sigmoid activation for latent disorder prediction in the context of mortality prediction models.

### Training mortality prediction models

Irrespective of the data encoding strategy or model architectures, all prediction models are trained using the same parameter setting. Binary cross-entropy is used as the loss function. Adam optimizer with a fixed learning rate of 0.001 and a batch size of 64 is used for training the models. Each model is trained to provide the best performance on the validation examples, and the best-performing model configuration is used for evaluating the test or held-out dataset.

### Training latent information prediction models

For training information leakage or latent prediction models, we again followed the same train, validation, and test split that is available for the prediction tasks. For estimating the information leakage from a trained model, we obtained the penultimate layer embedding for all examples. These embeddings are used as input representations for training and evaluating the latent information prediction models, i.e., gender, ethnicity, and disorder prediction models. Binary cross-entropy loss, Adam optimizer with a fixed learning rate of 0.001, and a batch size of 256 are used for training the models.

### Implementation details

All experiments are performed using Python. PyTorch is used as a deep-learning library. Quantum operations have been simulated using PennyLane[42]. Mutual information for the IB analysis (Fig. 6) has been estimated using ref. 43.

### Reporting summary

Further information on research design is available in the Nature Portfolio Reporting Summary linked to this article.

## Data availability

All datasets used in this study are publicly available from http://physionet.org/. MIMIC-III is available at https://physionet.org/content/mimiciii/1.4/. eICU-CRD is available at https://physionet.org/content/eicu-crd/2.0/. PhysioNet 2012 data is available at https://physionet.org/content/challenge-2012/1.0.0/. A PhysioNet account is required to access the datasets. Source data are provided in this paper.

## Code availability

The code repository is publicly available[44] and can be found at https://github.com/AnshThakur/Quantum-Encoding.

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

## Acknowledgements
D.A.C. was supported by the Pandemic Sciences Institute at the University of Oxford; the National Institute for Health Research (NIHR) Oxford Biomedical Research Center (BRC); an NIHR Research Professorship; a Royal Academy of Engineering Research Chair; the Wellcome Trust funded VITAL project; the UKRI; and the InnoHK Hong Kong Center for Center for Cerebro-cardiovascular Engineering (COCHE). T.Z. was supported by the Royal Academy of Engineering under the Research Fellowship scheme.

## Author contributions
A.T. and D.A.C. were responsible for developing the idea of data encoding for data democratization. A.T. and V.A. implemented the proposed encoding framework including random projections and random quantum encoding. A.T. and T.Z. studied the latent information leakage from the clinical models. A.T., J.A., and Y.W. designed the experiments and evaluated the impact of data encoding. A.T., V.A., T.Z., and D.A.C. were responsible for drafting the manuscript.

## Competing interests
The authors declare no competing interests.
