## [Peer Review File · Nature Communications]

Data Encoding For Healthcare Data Democratisation and Information Leakage PreventionREVIEWER COMMENTS

Reviewer #1 (Remarks to the Author):

The authors propose an approach to address the data democratization and latent information leakage in deep learning for health care. Specifically, an encoding framework is proposed where a multivariate time series is transformed into a new space that preserves the semantic information while at the same time making it challenging to reconstruct the original data. Two transformations are explored, random projections and random quantum circuits, that map input data to the same dimension by mixing across the dimensions. Results are presented on real world health care datasets and leading deep learning architecture such as TCN and ViT.

I think that the paper addresses an important challenge in healthcare where strict privacy requirements have impeded the development of foundational models and large multi-institution data releases. The proposed approach explores the problem in an innovative way by obfuscating the input time series in such a way that the trained models do not capture the sensitive characteristics from the data.

However, I have several concerns around guarantees and empirical evaluation. First, authors do not provide theoretical analysis on either of the two encoding methods and their properties. In a highly sensitive area such as patient privacy, I think that stronger guarantees are needed as any information leakage can have disastrous consequences.

Second, if an empirical approach is taken to demonstrate method efficacy, I would have liked to see more complex and deeper models evaluated. Transformers with one attention layer are unlikely to memorise a significant enough amount of data to fully evaluate latent information leakage, and it's been shown in NLP that deeper and larger models get increasingly better at data memorization. This also applies to the latent information prediction models. I couldn't find the details on model architecture in Section 4.4 but I'm assuming that a shallow MLP was used to predict the sensitive variables. This again raises the question of whether the performance can be improved with deeper and more complex classifiers.

Third, I think the random projection drops the accuracy too much to be useful, Figure 2 shows AUC drops of over 20 points for some models. The quantum circuit does better but is also more predictive for sensitive variables. For example Figure 3.c shows over 0.7 AUC for gender prediction with the Transformer model, is that too high given the sensitivity of the application?

Lastly, the authors say that their approach also addresses data democratization but this claim is not supported empirically. I think experiments predicting the sensitive variables directly from the features need to be added to verify this. Practically speaking, the data can only be shared if the sensitive information can not be inferred from it with sufficient accuracy.

Reviewer #2 (Remarks to the Author):

The manuscript entitled “Data Encoding for Healthcare Data Democratization and Information Leakage Prevention” by Thakur et al. describes two approaches to data encoding for training deep learning-based models for healthcare applications. The authors stress the importance of not violating privacy constraints in the healthcare sector, by encoding training data sets using an irreversible process that ultimately allows one to publicly share the encoded data. The manuscript is well written and presents the findings, based on an empirical study, in a convincing way. However, I find the discussion of the presented “quantum encoding” method slightly questionable. Therefore, I can only recommend the publication of the current manuscript in Nature Comms., if the following concerns are addressed:

[1] My main concern is about the usefulness of the “quantum encoding” scheme. The authors compare the performance of the quantum encoding scheme to a scheme based on random projections and find the quantum scheme to cause less deformation to the data, which they argue is beneficial for their application.

Is it really required to use a quantum circuit for this purpose, and should one not rather focus on solving this problem classically? The authors comment on this aspect on page 20 of the manuscript, where they state that studying non-linear data transformations could allow

one in the future to control the degree of deformation to the data. Such a solution would make much more sense to me and, in my opinion, enhance the quality of the work significantly.

[2] The authors use four qubits to implement the encoding circuit shown in Fig. 1(c). What is the reasoning behind choosing this many qubits? Will the deformation of the data increase when using more qubits? Can the authors envision a scenario where an actual quantum computer must be used for executing the encoding circuit?

[3] What is the motivation behind using the particular quantum circuit shown in Fig. 1 (c)? Does it encode the input data into the Hilbert space efficiently?

[4] All axis labels in Fig. 8 are missing. Without them this figure is very difficult to read.

[5] What is causing the periodicity of signals encoded with the random quantum circuit in Fig. 9?

Reviewer #3 (Remarks to the Author):

The paper introduces an encoding framework for healthcare data in order to reduce information leakage. The paper is very well written. The topic is relevant and interesting. I only have minor suggestions: Please add some practical implications of your results. I was wondering in Fig. 3 whether they also studied the difference in prediction for the different genders. Regarding Fig. 4 : did you selected these 4 ethnics or are these only and all ethnics covered by the chosen data sets? In terms of structuring, personally, I find it not really intuitive to read about the results before knowing the methodology. But I am assuming that this structure is given by the Nature template. If not, I would recommend to keep the traditional ordering. Methodology is sound.

Response to the Reviewers' Comments

Data Encoding For Healthcare Data Democratisation and Information Leakage Prevention

The authors thank the reviewers for their valuable comments. The authors have endeavoured to address all points raised by the reviewers. The paper has been revised to incorporate all suggestions and in the process, the authors believe that the manuscript has been significantly improved.

All the changes or newly added lines are coloured red in both manuscript and in the supplementary document. Sections 4 and 5 of this document highlight the major changes in the manuscript as well as the supplementary document.

1 Reviewer #1

1. **The authors propose an approach to address the data democratization and latent information leakage in deep learning for health care. Specifically, an encoding framework is proposed where a multivariate time series is transformed into a new space that preserves the semantic information while at the same time making it challenging to reconstruct the original data. Two transformations are explored, random projections and random quantum circuits, that map input data to the same dimension by mixing across the dimensions. Results are presented on real world health care datasets and leading deep learning architecture such as TCN and ViT. I think that the paper addresses an important challenge in healthcare where strict privacy requirements have impeded the development of foundational models and large multi-institution data releases. The proposed approach explores the problem in an innovative way by obfuscating the input time series in such a way that the trained models do not capture the sensitive characteristics from the data. However, I have several concerns around guarantees and empirical evaluation. *First, authors do not provide theoretical analysis on either of the two encoding methods and their properties. In a highly sensitive area such as patient privacy, I think that stronger guarantees are needed as any information leakage can have disastrous consequences.***

Reply: The authors thank the reviewer for their insightful comments, and would like to state that the main aim of this paper is to highlight and bring attention towards the potentials of data encoding in healthcare. The two data transformations used in the proposed framework were selected to showcase this potential as well as to argue for the invention of novel data transformations catering to the requirements of the proposed framework (*Section 1 i.e. Introduction* of the manuscript).

In the manuscript, we have highlighted that Random Projection are not irreversible and hence, are not ideal from privacy standpoint. However, the theoretical irreversibility of random quantum transformations makes them a viable data transformation candidate in the proposed framework. The random quantum entanglement operations and quantum measurements (that change the qubit states, irreversibly disrupts any quantum superposition and is many-to-one operation) makes sure that obtaining the original data from the encoded one is computationally near unfeasible. This information is already present in *Section 3 i.e. Discussion* of the manuscript

The scope of this manuscript was limited to empirical evaluation, which we have further revised as per the reviewer's comments to highlight that the proposed framework minimises potential sensitive information extraction.

(a) Training dynamics of transformers trained on MIMIC-III examples.

(b) Training dynamics of transformers trained on quantum-encoded MIMIC-III examples.

(c) Training dynamics of transformers trained on MIMIC-III examples encoded using random projections.

Figure R1: Training dynamics depicting the overfitting behaviour in deeper Transformers.

2. Second, if an empirical approach is taken to demonstrate method efficacy, I would have liked to see more complex and deeper models evaluated. Transformers with one attention layer are unlikely to memorise a significant enough amount of data to fully evaluate latent information leakage, and it's been shown in NLP that deeper and larger models get increasingly better at data memorization. This also applies to the latent information prediction models. I couldn't find the details on model architecture in Section 4.4 but I'm assuming that a shallow MLP was used to predict the sensitive variables. This again raises the question of whether the performance can be improved with deeper and more complex classifiers.

Reply: The authors thank the reviewer for this comment, and we would like to take this opportunity to clarify the following:

- **Deeper complex models:** Signals such as audio, images and text require deeper complex models for extracting semantically rich feature representations from inputs. However, the EHR data (non-textual), either tabular or time-series representing hospital stays, already contain features that can be considered

semantically rich and might not require the same level of information processing as required by other data types. Hence, the larger or complex models often over-fit on tabular and time-series EHR data resulting in potentially lesser generalisation (both on target task as well as on the latent prediction tasks), i.e. data memorised from training doesn’t “equate” with the better performance or better *information retrieval* in validation or test data.

This overfitting in deeper models is evident in Figure R1 that illustrates the training dynamics of transformers with 1, 2, 4 and 8 attention layers trained for mortality prediction on raw as well as encoded MIMIC-III datasets. Please note that barring the number of attention modules, the rest of architectural and training details are kept same.

The authors would further like to highlight that the number of attention layers in Transformers and ViT (as well as the number of layers in temporal convolutional networks) were considered as hyperparameters that were tuned to achieve best validation performance on the target tasks. However, we have repeated the latent gender prediction experiments with larger transformers and ViT as suggested by the reviewer. We have used early stopping and model checkpoints to obtain the best performing model configuration on validation examples. In this experiment, we have used the same one hidden layered MLP for latent prediction as used in the main manuscript (experiments with complex MLPs are discussed later).

Table 1: Effect of the number of attention layers in Transformers trained for mortality prediction on MIMIC-III dataset.

Depth	#Parameters (millions)	Encoding					
		No Encoding		Quantum		Random Projection	
		Mortality	Gender	Mortality	Gender	Mortality	Gender
1	1.25	0.838 ± 0.001	0.709 ± 0.002	0.816 ± 0.002	0.551 ± 0.001	0.719 ± 0.001	0.523 ± 0.002
2	2.51	0.831 ± 0.003	0.704 ± 0.002	0.811 ± 0.003	0.552 ± 0.002	0.718 ± 0.002	0.523 ± 0.001
4	5.01	0.826 ± 0.003	0.7 ± 0.001	0.813 ± 0.002	0.549 ± 0.001	0.717 ± 0.002	0.522 ± 0.002
8	10.1	0.827 ± 0.002	0.697 ± 0.002	0.812 ± 0.002	0.55 ± 0.002	0.717 ± 0.001	0.521 ± 0.002
16	20.21	0.826 ± 0.002	0.699 ± 0.002	0.813 ± 0.002	0.551 ± 0.001	0.717 ± 0.002	0.522 ± 0.001

Table 2: Effect of the number of attention layers in Transformers trained for mortality prediction on PhysioNet dataset.

Depth	#Parameters (millions)	Encoding					
		No Encoding		Quantum		Random Projection	
		Mortality	Gender	Mortality	Gender	Mortality	Gender
1	1.25	0.833 ± 0.002	0.756 ± 0.002	0.772 ± 0.003	0.578 ± 0.002	0.689 ± 0.003	0.527 ± 0.002
2	2.51	0.833 ± 0.003	0.755 ± 0.002	0.773 ± 0.002	0.577 ± 0.003	0.687 ± 0.001	0.527 ± 0.001
4	5.01	0.829 ± 0.002	0.753 ± 0.002	0.772 ± 0.001	0.575 ± 0.002	0.687 ± 0.002	0.526 ± 0.003
8	10.1	0.828 ± 0.001	0.754 ± 0.001	0.771 ± 0.003	0.576 ± 0.003	0.688 ± 0.002	0.525 ± 0.001
16	20.21	0.829 ± 0.001	0.753 ± 0.002	0.772 ± 0.002	0.576 ± 0.002	0.688 ± 0.003	0.526 ± 0.001

Table 3: Effect of the number of attention layers in Transformers trained for acute respiratory failure (ARF) prediction on the eICU dataset.

Depth	#Parameters (millions)	Encoding					
		No Encoding		Quantum		Random Projection	
		Mortality	Gender	Mortality	Gender	Mortality	Gender
1	1.25	0.766 ± 0.004	0.908 ± 0.003	0.753 ± 0.002	0.701 ± 0.003	0.698 ± 0.003	0.589 ± 0.004
2	2.51	0.762 ± 0.004	0.904 ± 0.005	0.756 ± 0.004	0.701 ± 0.004	0.695 ± 0.004	0.591 ± 0.003
4	5.01	0.765 ± 0.003	0.899 ± 0.005	0.754 ± 0.004	0.698 ± 0.004	0.691 ± 0.004	0.587 ± 0.005
8	10.1	0.763 ± 0.004	0.905 ± 0.003	0.751 ± 0.004	0.7 ± 0.005	0.696 ± 0.005	0.59 ± 0.003
16	20.21	0.765 ± 0.003	0.906 ± 0.004	0.755 ± 0.004	0.697 ± 0.004	0.699 ± 0.004	0.588 ± 0.003

Table 4: Effect of the number of attention layers in ViT trained for mortality prediction on MIMIC-III dataset.

Depth	#Parameters (millions)	Encoding					
		No Encoding		Quantum		Random Projection	
		Mortality	Gender	Mortality	Gender	Mortality	Gender
1	1.25	0.833 ± 0.003	0.689 ± 0.003	0.772 ± 0.003	0.546 ± 0.001	0.714 ± 0.004	0.493 ± 0.002
2	2.51	0.832 ± 0.003	0.689 ± 0.005	0.77 ± 0.004	0.547 ± 0.002	0.711 ± 0.002	0.493 ± 0.003
4	5.01	0.828 ± 0.004	0.688 ± 0.003	0.769 ± 0.003	0.545 ± 0.004	0.709 ± 0.003	0.489 ± 0.003
8	10.1	0.829 ± 0.004	0.687 ± 0.003	0.769 ± 0.003	0.547 ± 0.003	0.71 ± 0.003	0.488 ± 0.002
16	20.21	0.826 ± 0.002	0.688 ± 0.002	0.767 ± 0.004	0.546 ± 0.003	0.712 ± 0.002	0.492 ± 0.003

Table 5: Effect of the number of attention layers in ViT trained for mortality prediction on PhysioNet dataset.

Depth	#Parameters (millions)	Encoding					
		No Encoding		Quantum		Random Projection	
		Mortality	Gender	Mortality	Gender	Mortality	Gender
1	1.25	0.837 ± 0.003	0.711 ± 0.003	0.769 ± 0.003	0.563 ± 0.003	0.653 ± 0.003	0.552 ± 0.003
2	2.51	0.836 ± 0.004	0.71 ± 0.003	0.766 ± 0.004	0.561 ± 0.004	0.653 ± 0.003	0.554 ± 0.001
4	5.01	0.836 ± 0.003	0.71 ± 0.004	0.767 ± 0.002	0.562 ± 0.004	0.651 ± 0.004	0.551 ± 0.003
8	10.1	0.833 ± 0.003	0.708 ± 0.003	0.765 ± 0.004	0.559 ± 0.002	0.648 ± 0.004	0.55 ± 0.004
16	20.21	0.834 ± 0.004	0.709 ± 0.004	0.762 ± 0.005	0.561 ± 0.003	0.649 ± 0.004	0.549 ± 0.003

Table 6: Effect of the number of attention layers in ViT trained for acute respiratory failure (ARF) prediction on the eICU dataset.

Depth	#Parameters (millions)	Encoding					
		No Encoding		Quantum		Random Projection	
		Mortality	Gender	Mortality	Gender	Mortality	Gender
1	1.25	0.769 ± 0.004	0.858 ± 0.002	0.739 ± 0.003	0.662 ± 0.003	0.657 ± 0.004	0.591 ± 0.004
2	2.51	0.771 ± 0.003	0.856 ± 0.004	0.738 ± 0.005	0.662 ± 0.005	0.659 ± 0.003	0.588 ± 0.005
4	5.01	0.766 ± 0.004	0.853 ± 0.003	0.738 ± 0.004	0.659 ± 0.003	0.658 ± 0.005	0.589 ± 0.004
8	10.1	0.767 ± 0.003	0.855 ± 0.003	0.736 ± 0.003	0.661 ± 0.004	0.655 ± 0.004	0.588 ± 0.003
16	20.21	0.765 ± 0.003	0.852 ± 0.003	0.735 ± 0.004	0.657 ± 0.003	0.657 ± 0.004	0.587 ± 0.004

The results for transformers on MIMIC, PhysioNet and eICU are documented in Tables 1, 2 and 3, respectively. The analysis of these tables shows that no significant improvement is observed on both targeted (mortality prediction) as well as the latent (gender) prediction tasks as we increase the number of attention layers. Moreover, in some cases, we noticed a minute drop as model complexity is increased. Similar trends are observed for ViT as documented in Tables 4, 5 and 6.

This shows that larger models are not always better in healthcare informatics (non-textual), and these observations are consistent with widespread use of “comparatively” smaller deep models in this domain.

We have added this experiment in the supplementary document.

- Model for latent prediction tasks:** The authors would like to state that the purpose of latent prediction tasks was to exhibit that encoding enforces the information bottleneck principle. So, we wanted to analyse the characteristics of penultimate layer embedding for latent prediction without any further processing. As a result, we have used MLP with one layer to map these penultimate embedding to the latent outcomes. To analyse impact of complexity of the latent prediction model, we perform latent gender prediction tasks on best performing Transformers and ViT trained on MIMIC-III, PhysioNet and eICU datasets (discussed above). We varied the number of hidden layers from 1 to 5, each having 128 nodes and followed by ReLU activations. Apart from the depth of MLP, we didn’t change any experimental setting described in the manuscript. Again, early stopping and model checkpoints were used to select the best performing model configuration on validation examples.

Table 7: Impact on MLP complexity on the latent gender prediction from MIMIC-III transformer.

Depth	MIMIC-III		
	NO ENCODING	QUANTUM	RANDOM PROJECTION
1	0.709 ± 0.003	0.551 ± 0.005	0.523 ± 0.004
2	0.711 ± 0.002	0.551 ± 0.003	0.521 ± 0.002
4	0.704 ± 0.004	0.55 ± 0.002	0.523 ± 0.003
5	0.706 ± 0.003	0.55 ± 0.003	0.522 ± 0.005

Table 8: Impact on MLP complexity on the latent gender prediction from PhysioNet transformer.

Depth	PHYSIONET		
	NO ENCODING	QUANTUM	RANDOM PROJECTION
1	0.756 ± 0.002	0.578 ± 0.002	0.589 ± 0.004
2	0.753 ± 0.004	0.577 ± 0.004	0.589 ± 0.002
4	0.754 ± 0.003	0.579 ± 0.003	0.585 ± 0.006
5	0.751 ± 0.003	0.574 ± 0.005	0.587 ± 0.003

Table 9: Impact on MLP complexity on the latent gender prediction from eICU transformer.

Depth	eICU		
	NO ENCODING	QUANTUM	RANDOM PROJECTION
1	0.908 ± 0.003	0.701 ± 0.003	0.589 ± 0.004
2	0.906 ± 0.004	0.689 ± 0.003	0.588 ± 0.003
4	0.905 ± 0.002	0.702 ± 0.002	0.589 ± 0.006
5	0.906 ± 0.003	0.688 ± 0.003	0.591 ± 0.002

Tables 7, 8 and 9 illustrate that no significant improvement in performance of latent prediction from Transformers is observed on increasing the number of hidden layers in these MLPs. Again, similar behaviour is observed for ViTs (Tables 10, 11 and 12). The general trends of encoded data being worse than the standard ones in retaining input characteristics are still maintained. Again, we would like to point out that in many cases, increasing the depth or complexity of MLPs resulted in overfitting and didn't translate to improved predictive performance.

We have also added this experiment to the supplementary document.

Table 10: Impact on MLP complexity on the latent gender prediction from MIMIC-III ViT.

Depth	MIMIC-III		
	NO ENCODING	QUANTUM	RANDOM PROJECTION
1	0.672 \pm 0.004	0.549 \pm 0.003	0.489 \pm 0.004
2	0.671 \pm 0.003	0.551 \pm 0.004	0.491 \pm 0.002
4	0.67 \pm 0.004	0.547 \pm 0.004	0.491 \pm 0.004
5	0.671 \pm 0.004	0.548 \pm 0.003	0.488 \pm 0.003

Table 11: Impact on MLP complexity on the latent gender prediction from PhysioNet ViT.

Depth	PHYSIONET		
	NO ENCODING	QUANTUM	RANDOM PROJECTION
1	0.719 \pm 0.004	0.559 \pm 0.003	0.554 \pm 0.005
2	0.716 \pm 0.003	0.559 \pm 0.003	0.552 \pm 0.003
4	0.718 \pm 0.004	0.556 \pm 0.003	0.556 \pm 0.006
5	0.717 \pm 0.005	0.557 \pm 0.004	0.554 \pm 0.003

Table 12: Impact on MLP complexity on the latent gender prediction from eICU ViT.

Depth	eICU		
	NO ENCODING	QUANTUM	RANDOM PROJECTION
1	0.873 \pm 0.004	0.663 \pm 0.004	0.591 \pm 0.002
2	0.873 \pm 0.003	0.661 \pm 0.003	0.59 \pm 0.003
4	0.871 \pm 0.004	0.662 \pm 0.002	0.588 \pm 0.006
5	0.87 \pm 0.005	0.662 \pm 0.004	0.589 \pm 0.003

3. Third, I think the random projection drops the accuracy too much to be useful, Figure 2 shows AUC drops of over 20 points for some models. The quantum circuit does better but is also more predictive for sensitive variables. For example Figure 3.c shows over 0.7 AUC for gender prediction with the Transformer model, is that too high given the sensitivity of the application?

Reply:

- The authors agree with the reviewer that Random Projection is not an ideal transformation for the hypothesized framework. As discussed earlier, we have argued the disadvantages of Random Projection in Section 3 i.e. Discussion of the manuscript, highlighting that it is an easily reversible transformation and fails to preserve semantics up-to an acceptable level.

The purpose of using random projections was to exhibit that mere deformation of data is not enough, and an ideal data transform should also preserve semantics and be computationally irreversible.

- The authors would like to point out that compared to MIMIC and PhysioNet, gender prediction performance is significantly higher for eICU dataset across all scenarios (both encoding and raw). On raw time-series, gender prediction performance is as high as 0.88 – 0.91 for LSTM and Transformer. This is due to the fact that some input features such as height, weight and admission wards are highly correlated with gender in this dataset. Despite this correlation, quantum encoding resulted in an AUROC of around 0.7, a relative drop of 23.1% for transformers. Moreover, across majority of latent prediction experiments, we have seen satisfying performance.

If we remove these features that are highly correlated with gender (these features are non-clinical), we observe a relative average drop of 5.9% in latent gender prediction tasks across all models for quantum encoding (from 0.7 to 0.671 on transformers) while witnessing little to no drop in ARF prediction.

4. **Lastly, the authors say that their approach also addresses data democratization but this claim is not supported empirically. I think experiments predicting the sensitive variables directly from the features need to be added to verify this. Practically speaking, the data can only be shared if the sensitive information can not be inferred from it with sufficient accuracy..**

Reply: The authors thank the reviewer for pointing this out. Although there is a strong association between accuracy of the latent prediction (that is already there in paper) and predictive performance from the raw features, we agree that the empirical evaluation highlighted by the reviewer is required.

The authors have performed this experiment, and results and experimental setup are described below:

Experimental Setup: The experimental design used for the patient-care prediction tasks i.e. mortality and acute respiratory failure prediction (in the main manuscript) is also used for predicting gender and ethnicity from raw features. Train-test splits and model architectures i.e. LSTM, ViT, Transformers, temporal convolutional network (TCN) and multi-branch TCN network, described in the manuscript, are also used here.

We train each model for 200 epochs using Adam optimiser with a learning rate of 0.0001, early stopping and a batch size of 64 samples. AUROC is used as the performance metric.

Results: Figures R2 and R3 illustrate how different models perform when predicting gender and ethnicity from the raw or encoded time series data. The analysis of these figures highlight the following:

- As with the latent gender and ethnicity prediction tasks (discussed in the manuscript), the time-series encoding also results in a significant drop in performance of models trained on encoded time-series samples for predicting gender and ethnicity. Across all models, random projection results in a relative drop of 26.03%, 32.5% and 33.33%, respectively on MIMIC-III, PhysioNet and eICU gender prediction tasks. Similarly, quantum encoding results in average relative drops of 13.7%, 24.1% and 22.9%, respectively. Similar trends are observed for ethnicity prediction tasks.
- The analysis of these results provide a strong evidence that time-series encoding makes it hard to infer sensitive characteristics that can readily be extracted from the raw time-series data. If we analyse these results in association with mortality and ARF prediction tasks as well as latent prediction tasks, it is evident the proposed encoding framework achieves the desired characteristics of preserving semantics as well as masking sensitive information to a large extent.

The manuscript has been updated to include this experiment.

(a) Models' performance on MIMIC-III.

(b) Aggregate MIMIC-III performance.

(c) Models' performance on PhysioNet.

(d) Aggregate PhysioNet performance.

(e) Models' performance on eICU.

(f) Aggregate eICU performance.

Figure R2: Performance of LSTM, vision transformer (ViT), transformer, temporal convolutional network (TCN) and multi-branch temporal convolutional network (Multi-TCN) for predicting gender from raw time-series examples in (a) MIMIC-III, (c) PhysioNet and (e) eICU datasets, respectively. Gender prediction performance as a function of encoding methods across different models on (b) MIMIC-III, (d) PhysioNet and (f) eICU, respectively.

(a) Predicting if a patient is African-American.

(b) Predicting if a patient is Asian.

(c) Predicting if a patient is Caucasian.

(d) Predicting if a patient is Hispanic.

Figure R3: Model-specific and aggregated (across all models) performance for the task of predicting patients' ethnicity from the encoded time-series examples.

2 Reviewer #2

1. My main concern is about the usefulness of the “quantum encoding” scheme. The authors compare the performance of the quantum encoding scheme to a scheme based on random projections and find the quantum scheme to cause less deformation to the data, which they argue is beneficial for their application. Is it really required to use a quantum circuit for this purpose, and should one not rather focus on solving this problem classically? The authors comment on this aspect on page 20 of the manuscript, where they state that studying non-linear data transformations could allow one in the future to control the degree of deformation to the data. Such a solution would make much more sense to me and, in my opinion, enhance the quality of the work significantly.

Reply: The authors thank the reviewer for this insight and would like to argue the following:

- The target of this paper is to envisage an encoding framework that results in a near-irreversible or one-way data transformations to make the resultant encoded data imperceptible while preserving the semantic characteristics for model training (Introduction of the manuscript).
- Quantum information processing is mainly used to achieve the theoretical irreversibility of the encoded data transformation i.e. obtaining original data from the encoded one. The random quantum entanglement operations and quantum measurements (that change the qubit states, irreversibly disrupts any quantum superposition and is many-to-one operation) makes sure that obtaining the original data from the encoded one is computationally challenging. In healthcare scenarios where privacy is paramount, this property of quantum information processing makes it an ideal data transformation candidate in the proposed framework. *Section 3 Discussions* of the manuscript has been revised to explicitly state the importance of quantum encoding in proposed framework.
- Most existing data transforms (linear or non-linear, including random projections) are not entirely irreversible. While these transforms may preserve the semantics and may also induce imperceptibility, the reversible nature makes them undesirable for the proposed framework. With regard to random projections, we have discussed this fact in *Section 3 Discussion* of the manuscript.

While the authors aim to come up or invent these new non-linear transforms (as discussion in manuscript and pointed out by the reviewer) that hit the trifecta of the above-mentioned requirements and possibly improve some of drawbacks of quantum encoding, existing transforms are not entirely suitable for the hypothesised framework. Hence, leaving quantum encoding as an ideal existing candidate.

2. The authors use four qubits to implement the encoding circuit shown in Fig. 1(c). What is the reasoning behind choosing this many qubits? Will the deformation of the data increase when using more qubits? Can the authors envision a scenario where an actual quantum computer must be used for executing the encoding circuit?

Reply: The authors would like to state that the aim of the proposed framework is to distort the local characteristics of the input signal while preserving its global characteristics. The input signal is segmented into chunks, and these chunks are distorted by data transforms. Hence, the number of qubits is decided by the length of each chunk.

Smaller the chunks, lesser is the data distortion. Hence, the length of chunks play an important role in balancing the data obscuration and semantic preservation. We considered this length or number of qubits as a hyperparameter with possible values $\{2, 4, 6 \text{ and } 8\}$, and chose a number providing maximum performance of target tasks and minimum on latent prediction tasks. Almost in every case, we found that chunks of length

Table 13: Performance of LSTMs on different datasets as a function of length of segments in quantum encoding.

SEGMENT LENGTH OR NUMBER OF QUBITS	MIMIC-III		PHYSIONET		eICU	
	MORTALITY	GENDER	MORTALITY	GENDER	ARF	GENDER
2	0.829 ± 0.011	0.649 ± 0.008	0.809 ± 0.009	0.665 ± 0.012	0.772 ± 0.008	0.687 ± 0.013
4	0.823 ± 0.015	0.546 ± 0.009	0.789 ± 0.015	0.587 ± 0.017	0.757 ± 0.012	0.649 ± 0.008
6	0.814 ± 0.013	0.544 ± 0.02	0.772 ± 0.014	0.585 ± 0.013	0.749 ± 0.01	0.643 ± 0.011
8	0.811 ± 0.016	0.543 ± 0.017	0.763 ± 0.012	0.585 ± 0.009	0.738 ± 0.01	0.642 ± 0.013

4 provide better performance. Table 13 documents the performance of LSTMs on MIMIC-III as a function of length of chunks. The analysis of this table highlights that with 4-length segments, we obtain maximum target task performance and minimum latent task accuracy. Hence, we have used 4-qubits.

As discussed earlier, the major reason for utilising quantum circuits is to obtain the irreversible data transformations. Along with improving the execution time, the executions of the simulated quantum operations on the quantum computer might be essential to achieve this irreversibility.

3. **What is the motivation behind using the particular quantum circuit shown in Fig. 1 (c)? Does it encode the input data into the Hilbert space efficiently?**

Reply: Each layer of the quantum circuit used in this work is chosen to make sure that every qubit is entangled with or dependent on every other qubit. This operation is designed to breakdown the local structure of the input segment while preserving the semantics. Each qubit state after these operations can be seen as “aggregation” of all qubits, hence, the semantics are preserved to some extent.

The two-layered design is chosen to further maximize the entanglement within the segment to ensure the distortion of local structure.

4. **All axis labels in Fig. 8 are missing. Without them this figure is very difficult to read.**

Reply: The authors thank the reviewer for pointing this out. We have updated this figure and caption to provide more context.

5. **What is causing the periodicity of signals encoded with the random quantum circuit in Fig. 9?**

Reply: The signals depicted in Fig. 9 are slowly varying signals (first column of Fig. 9). Since we divide a signal into fixed-length segments of size N where N is the number of qubits (as described in Methods), the segments are almost similar with subtle differences due to slow-varying nature of the signal. Hence, each segment or chunk results in a similar encoded segmented as each chunk is processed by the same random quantum circuit. On concatenation, the encoded signal looks like a periodic signal.

A minor correction: The encoded signals illustrated in Figure 9 of the manuscript were obtained using the quantum circuit with 8 qubits. This figure has been revised to depict the encoded signals with the 4-qubit circuit. Note that the nature of encoded signals appears to be similar.

3 Reviewer #3

The authors thank the reviewer for their encouraging comment.

Table 14: Difference in mortality prediction as a function of gender. LSTM models with 256 units are used in this experiment. The average AUROC across 5 different runs is used as the performance metric.

Dataset	Encoding	Performance on Test Examples		
		Male	Female	All
MIMIC	NO ENCODING	0.8306	0.8612	0.8491
	RANDOM PROJECTION	0.6987	0.7265	0.7185
	QUANTUM	0.8026	0.8177	0.8116
PHYSIONET	NO ENCODING	0.8421	0.8507	0.8466
	RANDOM PROJECTION	0.5922	0.6011	0.5972
	QUANTUM	0.7847	0.7921	0.7871
EICU	NO ENCODING	0.7831	0.7684	0.7726
	RANDOM PROJECTION	0.6471	0.6378	0.6412
	QUANTUM	0.7572	0.7486	0.7525

1. **I only have minor suggestions: Please add some practical implications of your results.**

Reply: The authors thank the reviewer for this comment. The authors would like to clarify that the practical implications of the results lie in the confirmation that the hypothesised encoding framework (Introduction of the manuscript) has potential to be a solution to the data democratisation problem. We are able to come-up with an encoding framework that deforms the data to ensure implicit information bottleneck while exhibiting acceptable prediction performance. Discussion of the manuscript has been revised to explicitly state these facts.

2. **I was wondering in Fig. 3 whether they also studied the difference in prediction for the different genders.**

Reply: The authors have studied the difference in prediction performance as a function of gender across encoded as well as the normal data. Across all scenarios, the mortality prediction has little association with gender. Moreover, this nature of this association doesn't exhibit any drastic changes in models trained on encoded data. Table 14 of this document confirms this behaviour.

3. **Regarding Fig. 4 : did you selected these 4 ethnics or are these only and all ethnics covered by the chosen data sets?**

Reply: The authors would like to clarify that this figure covers four out of the five ethnicity categories present in this dataset. The "*Others*" ethnicity category is not considered in this experiment as it doesn't reveal specific or precise patient ethnicity.

4. **In terms of structuring, personally, I find it not really intuitive to read about the results before knowing the methodology. But I am assuming that this structure is given by the Nature template. If not, I would recommend to keep the traditional ordering. Methodology is sound.**

Reply: The authors would like to clarify that the current article structure is determined by the journal itself (Results and Discussion before Methods).

4 List of changes in the manuscript

1. A new experiment to predict the gender and ethnicity from the encoded signals has been added as Section 2.5 in the revised manuscript.
2. Figure 8 has been revised to add axis labels.
3. Figure 9 has been revised to illustrate the encoded signals with 4 qubit circuit.
4. Section 3 (Discussion) has been revised to describe why quantum encoding presents an ideal solution for data transformation in the proposed framework.

5 List of changes in the supplementary document

1. Experiments describing the relationship between the depth of models (transformers and ViT) and latent prediction performance has been added.
2. An experiment with the complex MLPs for latent prediction tasks has been added.

REVIEWERS' COMMENTS

Reviewer #1 (Remarks to the Author):

I wanted to thank the authors for a detailed response and additional experiments.

To clarify, my comments regarding larger models and theoretical analysis were aimed at the foundational architectures. If the data transformation approach proposed by the authors is accepted, it is likely that multiple potentially large encoded EHR datasets will be released. At the same time, recent research is increasingly showing that it is possible to build very large foundational models on tabular time-series data that do not overfit and generalise well to a number of downstream tasks (e.g. TimeGPT) analogous to NLP and vision. Self-supervision is key to training such models as purely supervised optimisation quickly leads to overfitting as authors demonstrate in Figure R1. While empirical evaluation of such methods is beyond the scope of this work, I think that the authors need to acknowledge (and discuss limitations of current work) that the encoded EHR data can and likely will be used in very large tabular models with hundreds of layers. So further analysis is needed to make sure that models of that capacity cannot extract sensitive information as that can have severe consequences.

In the fully supervised setting the additional experiments added by the authors show that increasing embedding model capacity, MLP capacity or predicting directly from transformed results does not lead to better classification accuracy for sensitive variables. These results are encouraging and satisfy my concerns for the empirical evaluation.

Stand-in for Reviewer 3 - I believe that all the concerns that were raised by Reviewer 3 are addressed in the rebuttal.

Reviewer #2 (Remarks to the Author):

I thank the authors for clarifying and addressing my main concerns in the revised version of the manuscript. I am happy to recommend the current version of the article for publication in Nature Communications.

Response to the Reviewers' Comments

Data Encoding For Healthcare Data Democratisation and Information Leakage Prevention

The authors thank the reviewers and the editor for their encouraging comments.

All the changes or newly added lines are coloured red in both manuscript and in the supplementary document. Sections 3 of this document highlights the major changes in the manuscript as per the reviewers' comments and editorial requirements.

1 Reviewer #1

1. I wanted to thank the authors for a detailed response and additional experiments.

To clarify, my comments regarding larger models and theoretical analysis were aimed at the foundational architectures. If the data transformation approach proposed by the authors is accepted, it is likely that multiple potentially large encoded EHR datasets will be released. At the same time, recent research is increasingly showing that it is possible to build very large foundational models on tabular time-series data that do not overfit and generalise well to a number of downstream tasks (e.g. TimeGPT) analogous to NLP and vision. Self-supervision is key to training such models as purely supervised optimisation quickly leads to overfitting as authors demonstrate in Figure R1. While empirical evaluation of such methods is beyond the scope of this work, I think that the authors need to acknowledge (and discuss limitations of current work) that the encoded EHR data can and likely will be used in very large tabular models with hundreds of layers. So further analysis is needed to make sure that models of that capacity cannot extract sensitive information as that can have severe consequences.

In the fully supervised setting the additional experiments added by the authors show that increasing embedding model capacity, MLP capacity or predicting directly from transformed results does not lead to better classification accuracy for sensitive variables. These results are encouraging and satisfy my concerns for the empirical evaluation.

Stand-in for Reviewer 3 - I believe that all the concerns that were raised by Reviewer 3 are addressed in the rebuttal.

Reply: The authors thank the reviewer for highlighting the issue of foundation models. We agree that very recent developments in this space are beyond the scope of the current work. This fact has been acknowledged in the Discussion section of the revised manuscript, positioning the absence of the study of foundation models as both a drawback and an area for future work.

2 Reviewer #2

1. I thank the authors for clarifying and addressing my main concerns in the revised version of the manuscript. I am happy to recommend the current version of the article for publication in

Nature Communications.

Reply: The authors express their gratitude to the reviewer for advocating the acceptance of the manuscript.

3 List of changes in the manuscript

1. Discussion has been revised to highlight that the analysis of foundation models is out of the scope of this manuscript.
2. Figure 6 from the previous manuscript, which illustrates the odds ratios of various acute and chronic conditions, has been moved to the supplementary document.
3. Figure 8 from the previous manuscript, which illustrates heat maps of encoded and raw time-series examples, has been moved to the supplementary document.
4. Figures 10 and 11 from the previous manuscript, which depict the performance of gender and ethnicity prediction from the raw datasets, have been moved to the supplementary document.